# Poor lie detection related to an under-reliance on statistical cues and overreliance on own behaviour
Sarah Ying Zheng [1,2,3] ✉, Liron Rozenkrantz[4,5] & Tali Sharot[3,4,6] ✉

The surge of online scams is taking a considerable financial and emotional toll. This is partially because humans are poor at detecting lies. In a series of three online experiments ($N_{exp1}$ = 102, $N_{exp2}$ = 108, $N_{exp3}$ = 100) where participants are given the opportunity to lie as well as to assess the potential lies of others, we show that poor lie detection is related to the suboptimal computations people engage in when assessing lies. Participants used their own lying behaviour to predict whether other people lied, despite this cue being uninformative, while under-using more predictive statistical cues. This was observed by comparing the weights participants assigned to different cues, to those of a model trained on the ground truth. Moreover, across individuals, reliance on statistical cues was associated with better discernment, while reliance on one's own behaviour was not. These findings suggest scam detection may be improved by using tools that augment relevant statistical cues.

Incidents of fraud have been escalating. In 2021, 2.8 million Americans lost a total sum of $5.8 billion to fraud—an increase of 70% from the previous year[1]. Most are committed online and include investment fraud, romance scams, false billing and phishing scams. Unfortunately, humans are not great at detecting lies, whether online or offline[2–11]. As a consequence, scams lead to significant emotional and financial damage every year[8,12].

Why are humans so poor at detecting lies (and thus fraud)? Answering this question is crucial, as identifying the cues that lead to suboptimal lie detection can inform the development of tools to help people detect and avoid scams. Yet, we have no precise understanding of what people are doing wrong. Here, we test the hypothesis that poor lie detection is partially explained by suboptimalities in the cues people use when assessing lies. In particular, relying on uninformative cues and underweighting more informative ones.

To try and understand *why* people are poor at detecting lies, we first need to consider *how* people go about detecting lies. Although little is known regarding the first question, a vast literature has focused on the second question, most notably in offline contexts (i.e., in person)[2–4,6,7,9,13–17]. These studies show people attempt to infer clues from others' facial expressions[13,14], eye and body movements[17] and voice[15]. Findings show these cues are generally not helpful[3,4,6,18] and even less so online (e.g., in relation to phishing e-mails, online chats, social media) where such cues typically are unavailable.

Suspicion, however, is not only driven by superficial cues such as facial expressions. 'Content cues' are crucial too[19,20]. For example, people assess the likelihood that a communication is true, given prior knowledge on the topic[19,20]. For instance, if a prospective date claims to be over 6'4", suspicion may arise, as only 5% or less of the population is of that height. When communication violates expectations due to statistical knowledge, a surprise signal will be triggered that may lead to suspicion. This mechanism is not to be confused with a distinct (but not mutually exclusive) proposal, according to which suspicion is generated based on the base rate of lies[21,22]. That is, in societies and contexts where dishonesty is more prevalent, people may generally be more suspicious[23].

It has also been suggested that people engage in theory of mind when assessing lies[20]. That is, they may consider other people's knowledge, feelings and motivations in assessing the likelihood of an utterance being a lie. For example, people consider whether a lie would benefit the liar[24–26]. Because people often use their own behaviour to understand and predict the behaviour of others[27–34], we predict they may also use their own behaviour as an informational cue to assess the honesty of others. That is, if an individual is dishonest in a certain context, e.g. on a dating site, that individual may believe others will likely be dishonest in that context, too.

Different types of informational cues may lead to conflicting assessments. For example, imagine someone claims to be a billionaire. On the one

[1]Department of Security & Crime Sciences, Faculty of Engineering, University College London, London, UK. [2]Dawes Centre for Future Crime, University College London, London, UK. [3]Department of Experimental Psychology, Faculty of Brain Sciences, University College London, London, UK. [4]Department of Brain and Cognitive Sciences, Massachusetts Institute of Technology, Cambridge, USA. [5]The Azrieli Faculty of Medicine, Bar-Ilan University, Tel Aviv, Israel. [6]The Max Planck UCL Centre for Computational Psychiatry and Ageing Research, University College London, London, UK. ✉e-mail: sarah.zheng@live.com; t.sharot@ucl.ac.uk

hand, because the assessor will not falsely claim to be a billionaire themselves, they may conclude others would not do so either and be *less* suspicious. On the other hand, the statistical likelihood of being a billionaire is low, which *increases* suspicion. Thus, these two types of assessments could drive suspicion in opposite directions. People's eventual suspicion of others would be the result of how they weigh these two possibly conflicting types of information. Some people may rely more on their own behaviour as a cue (i.e., self-projection), while others may rely more on statistical likelihood estimations and as a consequence arrive at different conclusions.

We hypothesise that individual differences in the weighting of different types of cues relates to how good people are at discerning honest from dishonest interactions. In particular, while relying on one's own behaviour to infer that of others likely requires less cognitive resources compared to assessing statistical likelihoods and inferring other people's motivations through theory of mind, it may be suboptimal, as inferences are made from just a single source (i.e., the self).

To test whether poor lie detection is associated with suboptimal weighting of informational cues, participants completed a controlled task similar in nature to many others used to study constructs such as (dis) honesty[20,24,26,35,36], altruism[36,37], morality[38,39] and cooperation[40–46]. The task was constructed such that participants could lie for their own monetary benefit at the expense of others. We quantified to what extent participants relied on their own behaviour, statistical likelihoods and others' inferred motives in assessing honesty. We then tested whether suboptimal lie detection was related to overreliance on one cue over another.

## Methods
### Participants Experiment 1
One-hundred-and-twenty participants were recruited through Prolific. The required sample size was determined based on a pilot study in which we found a Pearson's correlation of 0.286 between participants' lie detection accuracy (d'-scores) and their tendency to lie. We needed 94 participants to find a similar effect with .8 statistical power. We added just over 20% to account for dropout rates. UK and US residents with fluent English language comprehension who had not participated in any pilot studies participated. Data from participants whose honesty ratings variance was near-zero (N = 3) were not included, as we could not fit meaningful models to data with no variance. Three participants who failed at least three out of the nine (33%≤) of the attention check trials and 15 who failed too many comprehension checks were excluded from analyses. This left 102 participants (mean age = 32.6 (SD = 11.4), 64.7% female). Four of these participants encountered a technical issue that prevented saving survey responses after completing the task. Thus, analyses that include demographics and psychological traits are based on data from 98 participants. All were compensated at an hourly rate of £7.50, with a £0.05 bonus payment for every point they won during the task. The task took around 25 min to complete. The study was approved by the UCL Research Ethics Committee and all participants gave informed consent to participate.

### Participants Experiment 2
One-hundred-and-ten participants were recruited through Prolific. The required sample size was calculated as in Experiment 1. All pre-screening filters and financial compensation were the same as in Experiment 1 and participants could not have participated in Experiment 1. One participant with zero variance in their honesty ratings and one participant who failed more than 33% of the attention checks were excluded. Thus, data of 108 participants (mean age = 32.0 (SD = 10.9), 71.3% female) were analysed. The study was approved by the UCL Research Ethics Committee and all participants gave informed consent to participate.

### Participants Experiment 3
One-hundred-and-three participants were recruited through Prolific. All pre-screening filters were the same as Experiment 1 and 2 and participants could not have participated in Experiment 1 or 2. They were compensated at a base rate of £9.00 per hour, as data collection was done more than a year after

Experiment 1 and 2 and Prolific's advised base rate had increased since. Participants were told that each point they won or lost was worth £5 in the 'high stakes' trials block and £0.01 in the 'low stakes' trials block. Three participants who submitted incomplete data were excluded. Thus, data from 100 participants (mean age = 38.1 (SD = 13.3), 66% female) were analysed. At the end of the study, some participants played an extra trial in which they simply guessed if a card was blue or red, which negatively impacted their bonus payment. We added this trial to balance out bonus payoffs, which would otherwise be unaffordable. The study was approved by the UCL Research Ethics Committee and all participants gave informed consent to participate.

### Task procedure
Participants performed the task online through a Vue.js web application hosted via Google Firebase. After giving consent to participate in the study, participants were presented with the main task instructions and answered seven multiple choice comprehension check questions (see below). Next, they completed three practice trials and then proceeded to the cards task.

The task consisted of randomly sampled responses from actual past participants, divided over three blocks of 30 trials each. That is, each participant received a unique set of 90 trials, where every trial could contain a different past participant's response. This was to mimic online fraud scenarios in which individuals need to make veracity judgments in one-shot interactions with people they have never communicated with before. For Experiment 3, we only used two blocks, thus 60 trials in total. On each trial, the participant and the other participant (i.e., the past participant) were presented with a set of seven cards. The participant and the partner saw the same cards. Each card was either red or blue. Each trial consisted of at least one card of either colour. After 0.5 s, one of the displayed cards was randomly highlighted with a yellow border around the card for 1 s. This was the computer's random card pick for the participant. Given the actual random process, the computer could pick a different card to highlight for the participant and the other participant on a given trial. Participants were then asked to report the colour of the card that was picked for them, by clicking on a red or blue card.

One second later, the participant's and the other participant's reported card colour were revealed. The trial outcome was shown ('YOU WIN', 'YOU LOSE' or 'IT'S A TIE') and indicated with a gold coin or red cross below each participant's card report if either of them won or lost, respectively. When two participants reported different card colours, the one who reported blue always won one point and red always lost one point. When both participants reported the same card colour, they tied (no points were gained or lost by either person). If participants wanted to maximise their monetary reward, they could always report the blue card, regardless of their random card pick.

Finally, participants rated how honest they thought the other participant was on a horizontally displayed 6-point scale ('Completely dishonest'. 'Dishonest', 'Somewhat dishonest', 'Somewhat honest', 'Honest', 'Completely honest') and proceeded to the next trial. From trial 10 onward, participants answered an attention check question that asked them to select one of the six honesty rating scale options after they submitted their honesty rating (e.g., 'Please select the third option from the left'), every ninth trial. Each participant thus received nine attention checks during the main task in Experiments 1 and 2 and six attention checks in Experiment 3.

In Experiment 1, we used randomly selected trials to show another participant's response from all the responses that were already collected. Because on average, participants lied 25% of the time, participants observed another person lie on around 25% of the trials. For the first twenty participants in Experiment 1, we randomly sampled responses from pilot studies with the same task that used either 60 or 90 trials (pooled $N_{participants} = 50$, pooled $N_{trials} = 4200$). After this, we continuously added participants' responses to the responses pool to sample from for every next batch of up to twenty participants throughout Experiments 1 and 2. For Experiment 3, we used the same trials that participants saw in Experiment 1.

To examine whether the same results would be found if participants observed lies half the time (i.e., a higher lie base rate), we fixed the proportion of trials with lies at 50% in Experiment 2. That is, since the total number of trials remained 90 in Experiment 2, we randomly sampled 45 trials from the

growing pool of past participant responses in which their randomly selected card colour was red and they reported blue and 45 trials in which they reported honestly (both red and blue).

In Experiment 3, we informed participants after the first block of a change in the reward structure. That is, depending on their block order, that every win or loss would equate to X amount of money, instead of the amount previously instructed for the first block. Note that participants never received feedback on the actual honesty of the other participants. The comprehension check questions at the start ascertained that participants understood the stochastic nature and reward structure of the task (see below). Participants were only informed about their own stakes.

At the end of the task, participants in Experiment 1 answered the Empathy Quotient (EQ) questionnaire[47], Cognitive Reflection Test (CRT)[48] with one additional item 'If you are in a race and you pass the person in second place, what place will you be in?'[49], Revised Green et al. Paranoid Thoughts Scale (R-GPTS)[50], 10-item Autism Questionnaire (AQ)[51] and demographics questions (age, gender, education level and twin status). In Experiment 2 and 3, participants only answered the CRT and demographics items. We did not collect information on race/ethnicity. The study concluded with debrief questions on how clear the task instructions were, how many other participants they thought they played with, study purpose, others' and their own goals in the task, whether they would perform the task again, their overall honesty during the task and how good they thought they are at detecting deceptions (see Supplementary Notes 1). A total of four attention check questions of the form 'Please select <response item>' with a list of response items were added throughout the questionnaires in Experiment 1. The full task can be accessed at https://cards-dd-game.web.app/. None of the studies were pre-registered.

## Comprehension check questions

All participants were asked the following seven comprehension check questions after reading the instructions (correct answers are underlined):

1. In each trial, you and the other player are presented with an identical set of seven cards.
   a. <u>True</u>
   b. False
2. The computer picks one card for you and the other player at random. Which of the below statements is true?
   a. The computer picks the same card for you and the other player
   b. The computer picks a different card for you and the other player.
   c. <u>The computer randomly picked a card for the other participant and randomly picks a card for you. Your card pick is therefore not necessarily the same as the other player's.</u>
3. Let's say you reported a blue card and the other participant reported red:
   a. You win £0.01. The other player wins nothing.
   b. The other player wins. You win nothing.
   c. <u>You win £0.01. The other player loses.</u>
   d. The other player wins. You lose £0.01.
4. Let's say you and the other participant both reported a blue card. What happens?
   a. Each person loses
   b. <u>Neither one wins or loses</u>
   c. Each person wins
5. Let's say you reported a red card and the other participant reported blue:
   a. You win £0.01. The other player wins nothing.
   b. The other player wins. You win nothing.
   c. You win £0.01. The other player loses.
   d. <u>The other player wins. You lose £0.01.</u>
6. Let's say you and the other participant both reported a red card. What happens?
   a. Each person loses
   b. <u>Neither one wins or loses</u>
   c. Each person wins

7. Each trial, you will see another participant's report from a previous study.
   a. <u>True</u>
   b. False

All questions had to be answered correctly to be allowed to continue the study. Participants had to answer correctly at least at the second attempt. Participants were allowed to review the instructions after a first failed attempt. If they failed again, they were instructed to discontinue the study.

## Statistical analyses

To investigate what determines participants' suspicion levels on a trial by trial basis, we used a Bayesian 'model averaging' procedure adapted from Freckleton (2011)[52]. We reverse-coded participants' honesty ratings to reflect suspicion ratings and transformed them to be of range 0–1. Then, we defined four cues in three categories that may be associated with suspicion:

### Self-projection

One's own lying behaviour: a binary cue of whether the participant themselves lied on the trial. If people relied on their own behaviour to drive their suspicion, we would expect greater suspicion levels when participants lied themselves.

### Statistical cues

Signed expectation violation: participants could use the cards in front of them to infer the likelihood that a certain card is the 'computer chosen card'. When the probability is high, they would not be surprised when the partner declares that card was selected. However, when the probability is low, they would be surprised. Such surprise can be quantified as the difference between the reported colour and the expected colour, which we call expectation violation. In the 'signed' version, suspicion increases when the expectation violation is in favour of the partner and decreases when it is not in favour of the partner. Mathematically, this is equal to the value of the reported colour (−1 for red, 1 for blue) minus the value of the expected colour. Expected value equals the reported colour value (1 or −1) times the probability of that colour. Signed expectation violation values will therefore run from −1 to 1. For example, if the participant (thus, also the other participant) observed 3 blue cards and 4 red and the other participant reported red, the signed expectation violation would be −1 – (−1*0.57) = −0.429. That is, the participant should not be suspicious.

Unsigned expectation violation: 'unsigned' expectation violations reflect general surprise, regardless of the valence of the outcome. In the above example, when a participant observes 3 blue cards, 4 red cards and the other participant reporting red, the unsigned expectation violation would be 1−4/7 = 0.429. Here, as the range of values will be between 0 and 1, the participant would not be very surprised and report less suspicion.

### Motivation-related

Losing the trial: a binary cue reflecting whether the participant lost on the trial. Since losing means that the other person wins, losing a trial may increase participants' suspicion that the other person deliberately tried to win by lying, instead of reporting honestly.

We defined 15 models based on all possible combinations of one up to four of the above cues. We fitted each of them to participants' suspicion ratings, using linear mixed-effects models with fixed and random intercepts and fixed and random slopes. Next, using Bayesian 'model averaging'[52], we summed the cue estimates across all models that included the respective cue after weighting them by the normalised model fit probabilities based on Bayesian Information Criterion (BIC).

We interpreted the significance of the weighted cues through their 95% confidence intervals (CIs). Cues with CIs including zero after rounding to three decimals were deemed non-significant. The cues that remained significant after this procedure were used to define an overall model of suspicion.

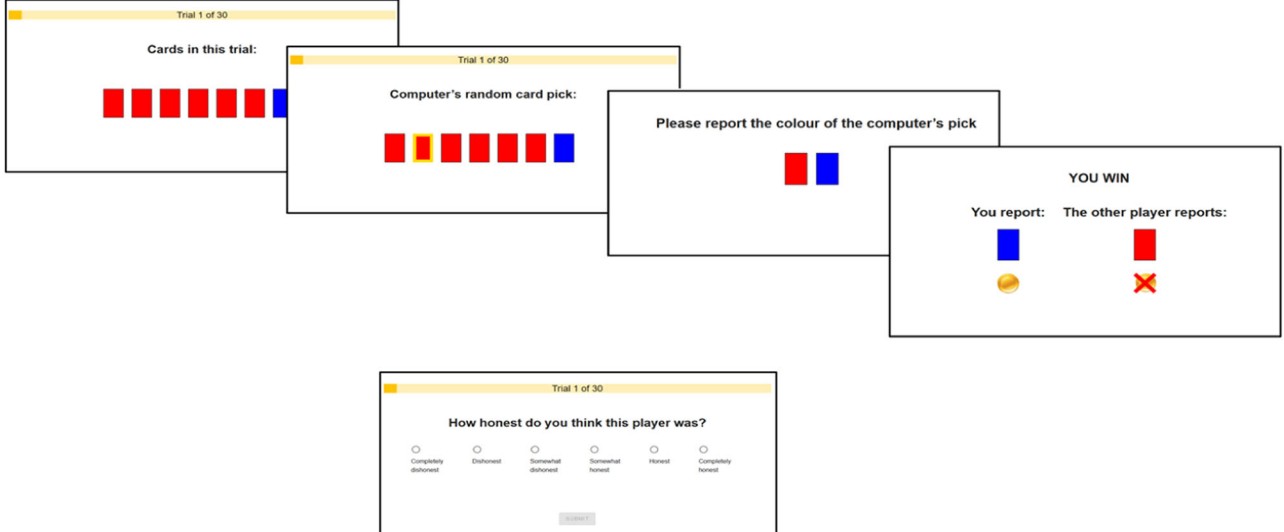

**Fig. 1 | Cards task.** On each trial, each participant is presented with a set of seven cards. The set is the same for the two participants, which they are aware off. Trials consisted of varying proportions of blue and red cards. Next, a yellow frame appears on one card, indicating that this card was randomly picked by the computer. The participant then needs to report whether this 'computer chosen card' was red or blue. A participant wins the trial if they report blue and the other participant reports red. They lose if they report red and the other participant reports blue, and tie if both report the same colour. Wins and losses are based on participants' reports, regardless of the ground truth. The participants' reports are then revealed, as well as the trial outcome (win, loss, tie). Finally, the participants rate the other participant's honesty. Participants do not perform the task with each other in real-time. Rather, they observe a previously gathered response for a specific trial, randomly sampled from all previous participants. Thus, each trial may be played against a different partner.

## Accurate lie detector

To see if the overall model of human suspicion reflects the best possible strategy to discern truths from lies, we compared it to the cues that drive suspicion of an accurate lie detector. In other words, by comparing the human suspicion model with that of an accurate detector's, we could examine whether the cues human participants used to assess honesty did in fact improve their judgements or perhaps impaired them. That is, is the use of these cues better than an accurate baseline model and would judgements be better if specific cues were not considered? In each study, we collated all participants' task data and applied the same model averaging procedure to predict this accurate lie detector's suspicion ratings, using linear regression. This detector's suspicion was 0 if the other participant did not lie and 1 if the other participant lied (i.e., reflecting the 'ground truth').

We also compared the accurate lie detector's reliance on the four cues with that of human participants, by examining whether the 95%-CIs for the weighted cue estimates of humans and the accurate detector overlapped. Non-overlapping CIs indicate a significant difference in how much humans and an accurate detector rely on a cue, overlapping CIs indicate no significant difference. We applied non-parametric tests for group-level comparisons involving participants' lying behaviour, as their tendency to lie followed a skewed distribution (see Supplementary Figs. 1c, 2c, 3c).

## Differences in lie detection accuracy

Our final goal was to see which individual differences may lead to more accurate honesty assessments. We first characterised individual differences in suspicion, by applying the same Bayesian averaging procedure on each participant's data, using linear regressions. From these, we obtained participants' weighted beta coefficients for each cue, which reflect their individual sensitivities to each of the suspicion cues we identified above. Next, to see what cues are associated with better discernment, we used the weighted beta coefficients in linear regressions predicting participants' lie detection accuracy, along with demographics, psychological traits and tendency to lie. For participants who never lied (Experiment 1 N = 16, Experiment 2 N = 13, Experiment 3 N = 27) and/or never lost a trial (Experiment 1 N = 9, Experiment 2 N = 6, Experiment 3 N = 13) the beta for the respective cue is zero.

Lie detection accuracy was computed as d'-scores from Signal Detection Theory (SDT)[53] to get an unbiased metric for how good participants actually were at discerning truths from lies. Since Experiments 1 and 3 only contain an average 25% of trials with lies, rating every response as 'honest' would be correct in 75% of the cases if measuring accuracy as the number of true positives and true negatives divided by the total number of cases to judge (i.e., overall accuracy). Measuring detection accuracy in terms of d'-scores is a more precise metric to indicate lie detection in contexts with imbalanced lie and truth base rates.

To compute d'-scores, participants' honesty ratings were first dichotomised by coding 'somewhat dishonest', 'dishonest' or 'completely dishonest' as 1 to reflect that they believed the other person lied on the given trial and 0 for the other ratings. Then, their d'-scores were computed according to the standard formula[53]:

$$d' = z(H) - z(FA), \qquad (1)$$

where hits (H) indicates the number of true positives (correctly detected lies) divided by the total number of trials on which the other participants lied, false alarms (FA) indicates the number of false positives (participant detected a lie, but the other participant did not actually lie) divided by the number of trials on which the other participants were honest and z indicates z-scoring.

In all analyses, statistical significance was inferred based on a two-tailed threshold probability of 0.05.

## Reporting summary

Further information on research design is available in the Nature Portfolio Reporting Summary linked to this article.

## Results
### Task overview

Participants completed an online cards task for two participants (Fig. 1). On every trial, each participant was presented with a set of seven cards. The presented set was the same for both participants. Each card was either red or blue and each trial consisted of varying card colour ratios (e.g., 6 red and 1 blue, 3 red and 3 blue etc.). Then, a yellow frame appeared on one of the

cards, indicating that the computer randomly and independently selected this card for the participant. This card could be the same or different for each participant. They then reported whether the chosen card was red or blue, by clicking on a red or blue card. Next, both card reports were revealed, as well as the trial outcome (win, loss, tie). A participant won if they reported blue and the other participant reported red. They lost if they reported red and the other participant reported blue, and tied if both reported the same colour. Wins or losses were based on the participants' reports, regardless of whether they were truthful. Participants did not perform the task with each other in real-time. Rather, participants observed responses randomly sampled from previous participants. Thus, each trial may be played with a different partner and participants were made aware of this (see Methods). All participants saw similar proportions of each card colour ratio and other participants' card reports (see Supplementary Figs. 1b, 2b, 3b).

At the end of each trial, participants rated how honest they thought the other participant was on a six-point Likert scale from 1 ('Completely dishonest') to 6 ('Completely honest'). The task consisted of three blocks of 30 trials each. Honesty ratings were reverse-coded and standardised to range between 0 and 1, such that 1 indicated high suspicion and 0 low suspicion (see Methods). To ensure participants fully understood the instructions they completed a comprehension check. Those who failed the check were not permitted to proceed to complete the task (see Methods for details).

## Experiment 1

One-hundred-and-two participants (mean age = 32.6 (SD = 11.4), 64.7% female) completed Experiment 1. They differed largely in how suspicious they were of others (mean averaged suspicion rating = 3.21, SD = 0.566; see Supplementary Fig. 1a) and how often they lied. On average, participants lied on 23.3% of the trials (M = 21, SD = 19.4 trials), with a third (32.4%) of participants rarely lying (in 0–5% of the trials) and a fifth (19.6%) lying on half or more of the trials (see Supplementary Fig. 1c).

## Human suspicion is explained by one's own behaviour, statistical likelihood computations and motivational cues

We first set out to test what cues are associated with suspicion within individuals on a trial-by-trial basis. To this end, we defined four candidate cues (based on the theories and literature described in the introduction) of different complexity that could drive suspicion levels (detailed below). We constructed models based on every possible combination of one up to all four cues and fitted these 15 models to participants' suspicion ratings, using linear mixed-effects models with fixed and random intercepts and fixed and random slopes. We then used Bayesian model averaging[52] to detect significant predictors of suspicion. Following this procedure, the standardised beta coefficients of each of the 15 models were weighted by the fitted model's normalised BIC (Bayesian Information Criterion). These weighted βs were then summed across all models that included the respective cues (see details in Methods). If the resulting 95% confidence interval (CI) of the weighted β of a cue included zero, it was deemed non-significant. This procedure revealed that all four cues significantly predicted participants' suspicion (see Fig. 2).

## Cue related to self-projection

(i) The first cue was whether the participant themselves lied on the trial. Participants were more suspicious of others when they lied themselves on a trial (mean weighted β = 0.043, 95%-CI = [0.033; 0.054]). This suggests that participants used their own behaviour as a proxy for that of others (i.e., self-projection).

## Cues related to statistical likelihood

(ii) The second cue—signed expectation violation—is associated with statistical likelihood and the other person's motivation to lie. Participants could use the cards in front of them to calculate the likelihood that a certain card is the 'chosen card'. Surprise can be quantified as the difference between the actual colour reported and the expected colour, which we call expectation violation. If the other participant declared a colour that indeed was likely to

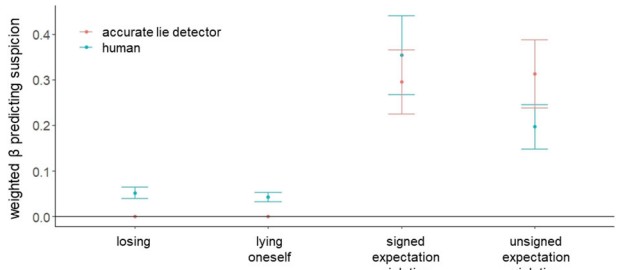

**Fig. 2 | Cues driving human suspicion as compared to an accurate lie detector.** We defined four candidate cues that could drive suspicion levels based on our main hypotheses: signed and unsigned expectation violation, lying oneself, and losing a trial. We constructed models based on every possible combination of one up to all four cues and fitted these 15 models to participants' suspicion ratings (N = 102), using linear mixed-effects models with fixed and random intercepts and fixed and random slopes. The standardised beta coefficients were then weighted by the fitted model's normalised BIC (Bayesian Information Criterion) and summed across all models that included the respective cue. This procedure revealed that all four cues significantly predicted suspicion in humans (turquoise). The same procedure was applied to the suspicion of a simulated, accurate lie detector (that is, an agent which would always be suspicious when someone lies and not otherwise). Suspicion of this accurate detector was associated only with signed and unsigned expectation violation, but not with information about participants' own behaviour or losing (red). Points represent weighted beta estimates. Whiskers depict the upper and lower bounds of the 95% confidence interval.

have been randomly chosen, expectation violation would be low (no surprise). However, if the probability that that colour would be chosen is low, expectation violation would be high (high surprise). This quantity is then 'signed', because suspicion should be high when the expectation violation is *in favour* of the other participant (that is, they have a motivation to lie) and negative otherwise. Mathematically, this is equal to the value of the other participant's reported card (−1 for red, 1 for blue) minus the value of the expected colour (−1 for red, 1 for blue) times the probability of the reported colour. For example, if the participant observed 3 blue cards and 4 red and the other participant reported *red*, the signed expectation violation would be equal to (−1) – (−1*4/7) = −0.429. When these values are high in magnitude and positive in sign, the surprise is expected to increase overall suspicion and vice versa. Indeed, participants were more suspicious of others when others reported a statistically surprising card that was in the others' favour (mean weighted β = 0.354, 95%-CI = [0.267; 0.44]).

(iii) The third cue, associated with statistical likelihood, was unsigned expectation violation, which reflects general surprise. It is a simpler version of signed expectation violation, as it disregards the valence of the reported card and merely considers the probability of observing the reported card colour. Thus, in the above example with 3 blue and 4 red cards and the other participant reporting red, the unsigned expectation violation would be 1−4/7 = 0.429. Participants were more suspicious of others when the other participant reported a generally surprising card (mean weighted β = 0.197, 95%-CI = [0.149; 0.245]).

## Cues related to potential lying motivation

(iv) The fourth cue, associated with potential lying motivation, was whether the participant lost on the trial. The logic behind this is that when a participant lost, they may infer that they lost because the other person lied. Indeed, when a participant lost, their suspicion increased (mean weighted β = 0.053, 95%-CI = [0.04; 0.066]).

The four cues were moderately correlated with each other (see Supplementary Table 1a), as they share some variance (e.g., a participant will be unlikely to lose on a trial in which they lied), yet incorporate a large amount of unshared variance that allowed the models to converge. We confirm these findings using a leave-one-out cross-validation procedure and also after excluding participants (6.9%) who did not believe they were playing with other human participants (see Supplementary Tables 2a, 3). Moreover, we

find the same results using a Brunswik lens model analysis[54] (see Supplementary Notes 3) and when considering a fifth cue for the other person's reported card colour (see Supplementary Table 5). The Brunswik lens model provides an alternative framework to analyse what information sources humans use versus an accurate lie detector in judging honesty.

## An accurate lie detector uses statistical likelihoods and not participants' own behaviour

Next, we examined which combination of the above cues best predicted the ground truth. To that end, we repeated the model averaging procedure exactly as described above on all trials, except that instead of the dependent variable (the Y) being participants' suspicion rating, it was the ground truth (i.e., a rating of 1 when the other participant lied and 0 when the other participant was honest). The independent variables (the Xs) were all exactly as above. We termed the resulting Bayesian averaged linear model an 'accurate lie detector' – it reflects the best combination of beta coefficients of the cues above to predict if someone lied or not. The resulting linear model revealed significant positive weights assigned to the cues of statistical nature: signed expectation violation (mean weighted $\beta = 0.295$, 95%-CI = [0.225; 0.365]) and unsigned expectation violation (mean weighted $\beta = 0.313$, 95%-CI = [0.238; 0.387]), but zero beta coefficients to the rest of the cues. Thus, in contrast to humans, the accurate lie detector model did not rely on participants' own lies (mean weighted $\beta = 0$, 95%-CI = [0; 0]) nor whether they lost (weighted $\beta = 0$, 95%-CI = [0; 0], see Fig. 2). We confirm these findings in a leave-one-out cross-validation analysis (see Supplementary Table 2b).

To examine whether humans may under- or over-weight certain cues relative to the accurate lie detector model, we examined whether the CIs of humans' betas and of the accurate lie detector overlapped. This showed that humans over-relied on self-projection (i.e., lying oneself) and losing relative to the accurate lie detector model, with no significant difference for signed expectation violation. Humans somewhat under-weight unsigned expectation violation relative to the accurate detector, however, the 95%-CIs overlapped slightly.

## Reliance on statistical likelihoods but not on one's own behaviour is related to better lie detection

After identifying what cues drive people's suspicion, we tested which of these are related to more accurate veracity judgements. First, we ran the same Bayesian averaging procedure explained above, but separately for each participant to obtain their weighted beta estimates for each of the four cues identified above. This provided us with beta coefficients for each participant (i.e., the weight a participant assigns to each cue when generating suspicion), which we then related to how accurate each participant was in their suspicion (for participants who never lied (N = 16) and never lost (N = 9) the respective betas are zero). The beta coefficients across all cues were significantly positive for most participants (range across cues: 62.8–89.2%; see Supplementary Table 4).

To quantify how good participants were at lie detection we first dichotomised their suspicion ratings to 1 if the rating corresponded to the 'dishonest' half of the rating scale, indicating that participants thought that the other person lied, or 0 otherwise. These values were used to obtain participants' false alarm rates (i.e., the proportion of trials incorrectly marked as dishonest) and hit rates (i.e., the proportion of lies that were indeed detected) to compute d'-scores according to Signal Detection Theory (SDT)[53]. These scores reflect participants' ability to discern honesty from lies, where higher scores indicate higher detection accuracy (see Methods for details). On average, participants had a d'-score of 0.937 (SD = 0.534). That is, they correctly identified 44.2% of the lies (M = 9.95, SD = 6) and wrongly suspected lies on 20.2% (M = 13.6, SD = 10.8) of the trials on average. We then ran a linear regression to predict participants' d'-scores from their individual beta coefficients, demographics (age, gender, education level), psychological traits (autism, empathy, cognitive reflection, paranoia) and how often they lied themselves.

We found that better lie detection (i.e., d'-scores) was predicted by beta coefficients relating suspicion to signed expectation violations (standardised

$\beta = 0.683$, t(89) = 8.51, $p < 0.001$, 95%-CI = [0.337; 1.03]), unsigned expectation violations (standardised $\beta = 0.593$, t(89) = 8.55, $p < 0.001$, 95%-CI = [0.255; 0.932]) and losing (standardised $\beta = 0.404$, t(89) = 5.28, $p < 0.001$, 95%-CI = [−0.04; 0.847]), but not by beta coefficients that related suspicion to one's own lies (standardised $\beta = −0.047$, t(89) = −0.701, $p = 0.485$, 95%-CI = [−0.928; 0.834]), nor participants' overall tendency to lie (standardised $\beta = −0.012$, t(89) = −0.167, $p = 0.868$, 95%-CI = [−0.23; 0.206]; Fig. 3a, b). No other factors were significant (age: standardised $\beta = −0.087$, t(89) = −1.19, $p = 0.237$, 95%-CI = [−0.094; −0.079]; gender: standardised $\beta = −0.119$, t(89) = −1.6, $p = 0.113$, 95%-CI = [−0.29; 0.051]; education level: standardised $\beta = 0.117$, t(89) = 1.72, $p = 0.09$, 95%-CI = [0.072; 0.162]; CRT score: standardised $\beta = 0.005$, t(89) = 0.06, $p = 0.95$, 95%-CI = [−0.062; 0.071]; EQ score: standardised $\beta = −0.02$, t(89) = −0.24, $p = 0.81$, 95%-CI = [−0.151; 0.112]; R-GPTS score: standardised $\beta = 0.073$, t(89) = 0.97, $p = 0.334$, 95%-CI = [−0.019; 0.165]; AQ score: standardised $\beta = 0.063$, t(89) = 0.75, $p = 0.456$, 95%-CI = [−0.478; 0.604]). Thus, relying on statistical likelihoods improves the ability to detect lies, but relying on one's own lying behaviour does not. Losing was also associated with discernment (i.e., d'-scores) because when the other participant lies, they do so in their favour, which increases the likelihood that the participant will lose. Similar results are observed when lie detection is calculated using different accuracy measures (see Supplementary Notes 2).

## An accurate lie detector outperforms humans

We computed the accurate lie detector model's d'-score for each set of trials a participant saw and compared it to that participant's d'-score. On average, the accurate lie detector had a d'-score of 1.44 (SD = 0.375), which was significantly better than humans (Wilcoxon signed rank test = 4922, $p < 0.001$; Fig. 3c). Since the accurate detector's model does not rely on information regarding participants' own lying behaviour and the human suspicion model does, this result suggests that humans are suboptimal at detecting lies, because they use information on their own behaviour.

## People who lie more are more suspicious of others

Lastly, we examined if the individual differences in suspicion were associated with the individual differences in lying. We ran a linear regression predicting participants' mean suspicion ratings from demographics (age, gender, education level), psychological traits (autism, empathy, cognitive reflection (CRT), paranoia) and tendency to lie (i.e., proportion of trials they lied themselves). We found that greater suspicion was associated with more lying (standardised $\beta = 0.268$, t(93) = 2.83, p = .006, 95%-CI = [−0.015; 0.55], Fig. 3d) and marginally with higher scores on the Revised Green et al. Paranoid Thoughts Scale (R-GPTS)[50] which measures paranoid tendencies (standardised $\beta = 0.21$, t(93) = 2.01, $p = 0.048$, 95%-CI = [0.085; 0.336]). No other factors were significant (age: standardised $\beta = −0.156$, t(93) = −1.53, $p = 0.131$, 95%-CI = [−0.166; −0.146]; gender: standardised $\beta = −0.118$, t(93) = −1.15, $p = 0.252$, 95%-CI = [−0.348; 0.111]; education level: standardised $\beta = 0.011$, t(93) = 0.11, $p = 0.913$, 95%-CI = [−0.051; 0.072]; CRT score: standardised $\beta = 0.091$, t(93) = 0.92, $p = 0.362$, 95%-CI = [0.005; 0.177]; EQ score: standardised $\beta = 0.205$, t(93) = 1.82, $p = 0.072$, 95%-CI = [0.027; 0.382]; AQ score: standardised $\beta = 0.144$, t(93) = 1.22, $p = 0.228$, 95%-CI = [−0.594; 0.881]). This suggests that participants who are more inclined to lie themselves are more suspicious of others, which also is in line with the idea that participants use their own lying to infer the honesty of others.

Thus far, we observed that participants who lied more were more suspicious of others. This finding may indicate that people use their own behaviour to infer the behaviour of others ('If I lie, others probably lie too') or that people's perception of others alters their own behaviour ('Others are lying, so I will too'), or both. To test these possibilities, we conducted two additional experiments. In Experiment 2, we manipulated the percentage of times others lied and examined if participants lied more. In Experiment 3, we manipulated participants' lying behaviour by altering their incentive to lie, without changing others' lying tendency. These two additional

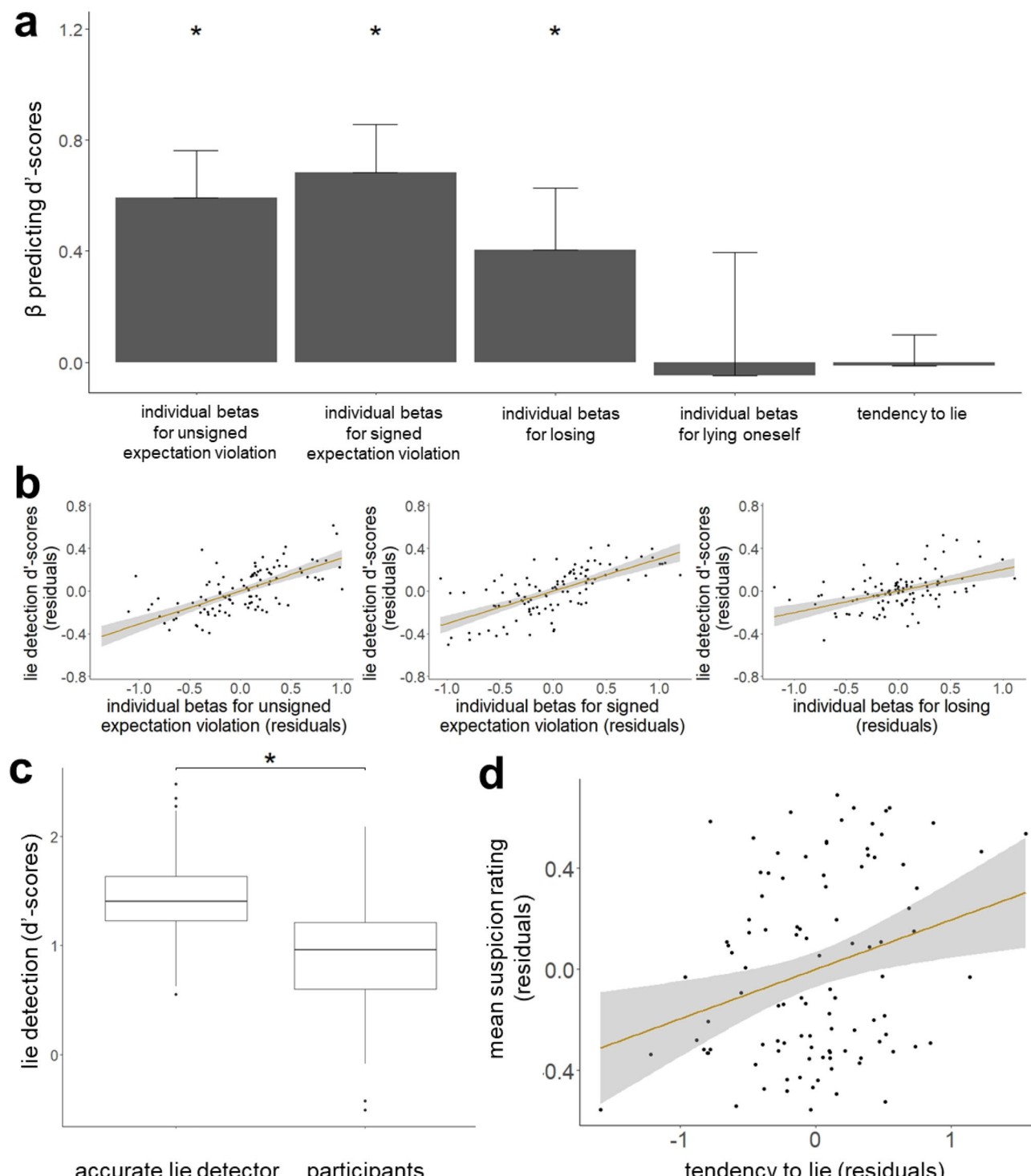

**Fig. 3 | Better lie detection is associated with reliance on statistical likelihoods and losing, but not with reliance on one's own lying. a** Better discernment (d'-scores) across individuals (N = 102) was related to greater weight assigned to signed expectation violations (standardised β = 0.683, t(89) = 8.51, $p < 0.001$, 95%-CI = [0.337; 1.03]), unsigned expectation violations (standardised β = 0.593, t(89) = 8.55, $p < 0.001$, 95%-CI = [0.255; 0.932]) and losing (standardised β = 0.404, t(89) = 5.28, $p < 0.001$, 95%-CI = [−0.04; 0.847]), but not by beta coefficients that related suspicion to one's own lies (standardised β = −0.047, t(89) = −0.701, $p = 0.485$, 95%-CI = [−0.928; 0.834]), nor participants' overall tendency to lie (standardised β = −0.012, t(89) = −0.167, $p = 0.868$, 95%-CI = [−0.23; 0.206]). Whiskers indicate standard error. * $p < 0.001$ **b** Partial regression showing that across individuals, greater beta estimates that relate suspicion to unsigned and signed expectation violations and losing are associated with discernment (d'-scores). Shaded areas indicate 95% confidence intervals. Dots represent individual participants. **c** An accurate lie detector that does not consider information on when participants lie themselves outperforms human lie detection (Wilcoxon signed rank test = 4922, $p < 0.001$, N = 102). Horizontal lines indicate median values, boxes indicate 25–75% interquartile range, whiskers indicate 1.5 × interquartile range. * $p < 0.001$ **d** People who lied more themselves were more suspicious of others. The relationship between tendency to lie (i.e. proportion of trials on which an individual lied) and mean suspicion rating is shown, controlling for all demographics and scores on psychological and cognitive trait questionnaires (standardised β = 0.276, t(89) = 2.882, $p = 0.005$, 95%-CI = [−0.015; 0.55]). Shaded area indicates 95% confidence interval. Dots represent individual participants.

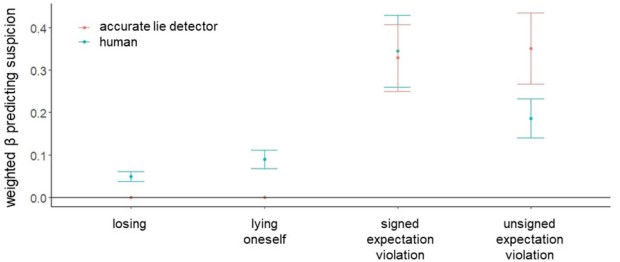

**Fig. 4 | Replication of cues driving human suspicion as compared to an accurate lie detector.** We constructed models based on every possible combination of one up to all four cues (signed and unsigned expectation violation, self-projection and losing). and fitted these 15 models to participants' suspicion ratings (N = 108), using linear mixed-effects models with fixed and random intercepts and fixed and random slopes. The standardised beta coefficients were then weighted by the fitted model's normalised BIC (Bayesian Information Criterion) and summed across all models that included the respective cue. This procedure revealed that all four cues significantly predicted suspicion in humans. The same procedure was applied to the suspicion of a simulated accurate lie detector (that is, an agent that would always be suspicious when someone lies and not otherwise). Suspicion of this simulated accurate lie detector was associated only with signed and unsigned expectation violation, but not with information about participants' own behaviour or losing. Points represent weighted beta estimates, whiskers depict the upper and lower bounds of the 95% confidence intervals (CIs).

experiments also enabled us to replicate the results of Experiment 1 under different variations of the task.

## Experiment 2
In Experiment 1, we randomly picked other participants' responses from all the responses we had collected to present to the 'live' participants. Because on average participants lied on ~25% of the trials, participants observed another person's lie ~25% of the time. To examine whether participants would lie more when observing more lies, we fixed the amount of trials with lies to 50%. We did not collect any questionnaire responses, except for the CRT, to validate the null result.

One-hundred-and-eight participants (mean age= 32.0 (SD = 10.9), 71.3% female) completed Experiment 2. On average, they were more suspicious of others than in Experiment 1 (M = 2.969, SD = 0.578, Mann-Whitney U = 7116, $p < 0.001$). This means the manipulation was successful, as others indeed lied more. Importantly, however, we found no statistically significant evidence for a difference in how often participants themselves lied in Experiment 2 (~28.3% of the trials, M = 25.5, SD = 21.5) compared to participants in Experiment 1 (Mann-Whitney U test = 5730.5, $p = 0.611$). Thus, participants were more suspicious of others, but this did not increase their own lying. Instead, greater suspicion was likely the result of higher signed and unsigned expectation violations in Experiment 2 compared to Experiment 1 (signed expectation violation: Mann-Whitney U = 10092, $p < 0.001$; unsigned expectation violation: Mann-Whitney U = 10875.5, $p < 0.001$) and participants in Experiment 2 lost more than in Experiment 1 (Mann-Whitney U = 6628.5, $p = 0.011$), because others lied more than they did. In other words, this result does not support the idea that the key finding in Experiment 1—an association between people's own lying and their suspicion of others – is because people's perception of others altered their own behaviour ('Others are lying, so I will too').

The model averaging procedure successfully replicated Experiment 1 (Fig. 4). Within individuals, suspicion was driven by the same four cues as in Experiment 1: suspicions increased on trials in which participants lied themselves (mean weighted β = 0.09, 95%-CI = [0.068; 0.112]), when others reported a statistically unlikely card colour which was in the other person's favour (i.e., signed expectation violation; mean weighted β = 0.345, 95%-CI = [0.26; 0.429]), when others' card reports were generally more surprising (i.e., unsigned expectation violation; mean weighted β = 0.186, 95%-CI = [0.141; 0.232]) and when participants lost (mean weighted β = 0.05, 95%-CI = [0.038; 0.062]). We

confirm these findings using a leave-one-out cross-validation procedure and also after excluding participants (15.7%) who did not believe they were playing with other human participants (see Supplementary Tables 2a, 3).

As in Experiment 1, we found that the suspicion of an accurate lie detector was associated with signed expectation violation (mean weighted β = 0.329, 95%-CI = [0.25; 0.408]) and general surprise (i.e., unsigned expectation violation; mean weighted β = 0.351, 95%-CI = [0.267; 0.435]), but not with whether participants lied (mean weighted β = 0, 95%-CI = [0; 0]), or lost (mean weighted β = 0, 95%-CI = [0; 0]). The non-overlapping CIs further indicate that humans rely on self-projection (i.e., their own lying behaviour) and losing more than the accurate lie detector, and rely less on unsigned expectation violation (Fig. 4).

Next, we investigated what cues were associated with better discernment. Participants had an average d'-score of 0.833 (SD = 0.482) and correctly identified 47.1% of the lies (M = 21.2, SD = 10.7) and wrongly suspected lies in 20.9% (M = 9.42, SD = 6.57) of the trials on average. Consistent with Experiment 1, we found that greater beta coefficients of signed expectation violation (standardised β = 0.73, t(98) = 9.25, $p < 0.001$, 95%-CI = [0.466; 0.994]), unsigned expectation violation (standardised β = 0.551, t(98) = 8.5, p < 0.001, 95%-CI = [0.228; 0.875]) and losing (standardised β = 0.248, t(98) = 3.31, $p = 0.001$, 95%-CI = [−0.159; 0.655]), but not participants' betas for their own lying (standardised β = −0.023, t(98) = −0.338, p = .736, 95%-CI = [−0.64; 0.594]), nor their tendency to lie (standardised β = −0.02, t(98) = −0.31, $p = 0.758$, 95%-CI = [−0.186; 0.146]), predicted better d'-scores. We also found an effect of gender, where female participants had lower d'-scores (standardised β = −0.158, t(98) = −2.46, $p = 0.016$, 95%-CI = [−0.304; −0.013]), though this was not observed in Experiment 1 (Fig. 5a, b). As in Experiment 1, the accurate detector's d'-scores were greater than humans' (M = 1.56, SD = 0.283; Wilcoxon signed rank test = 5595, $p < 0.001$; Fig. 5c). These results further corroborate that using statistical likelihood estimation and not one's own behaviour, improves lie detection.

Consistent with Experiment 1, we found that people who lie more also believe others lie more (standardised β = 0.364, t(102) = 3.98, $p < 0.001$, 95%-CI = [0.099; 0.63]; Fig. 5d). This aligns with our conclusion that people infer others' honesty by projecting their own behaviour onto others.

## Experiment 3
Experiment 2 showed that participants who believe that other people are lying were not necessarily inclined to lie more themselves. Here, we ask if the reverse may be true—does lying more oneself lead one to believe others lie as well? To that end, we performed a third experiment in which we manipulated the likelihood that a participant would lie, by altering their incentive to do so without changing the frequency by which others lie.

Specifically, one-hundred participants (mean age = 38.1 (SD = 13.3), 66% female) completed two blocks of trials from Experiment 1 in which others lied ~25% of the time. However, this time, £5 were at stake on each trial in one block (high stakes), while in the other block only £0.01 was at stake (low stakes, counterbalanced order). The question was whether higher stakes would lead participants to lie more and as a result become more suspicious of others, despite the fact that others' tendency to lie remained constant over the blocks (i.e., each participant saw the same responses from others as the participants in Experiment 1).

Participants lied more in the high stakes block than in the low stakes block (Wilcoxon signed rank test = 379.5, $p = 0.002$; Fig. 6a), suggesting the manipulation worked. Importantly, participants were also more suspicious of others in the high stakes block than in the low stakes block (Wilcoxon signed rank test = 1306, $p = 0.002$; Fig. 6b), despite that the amount of lies of the other participants was similar across blocks.

It is possible that participants' suspicion increased in the high stakes block solely because they assumed other participants completed the task under the same conditions (which was not the actual case) and not because they themselves lied more. To test this, we performed a mediation analysis to examine if the relationship between blocks and suspicion is mediated by participants' own lying. We found that suspicion was associated with block

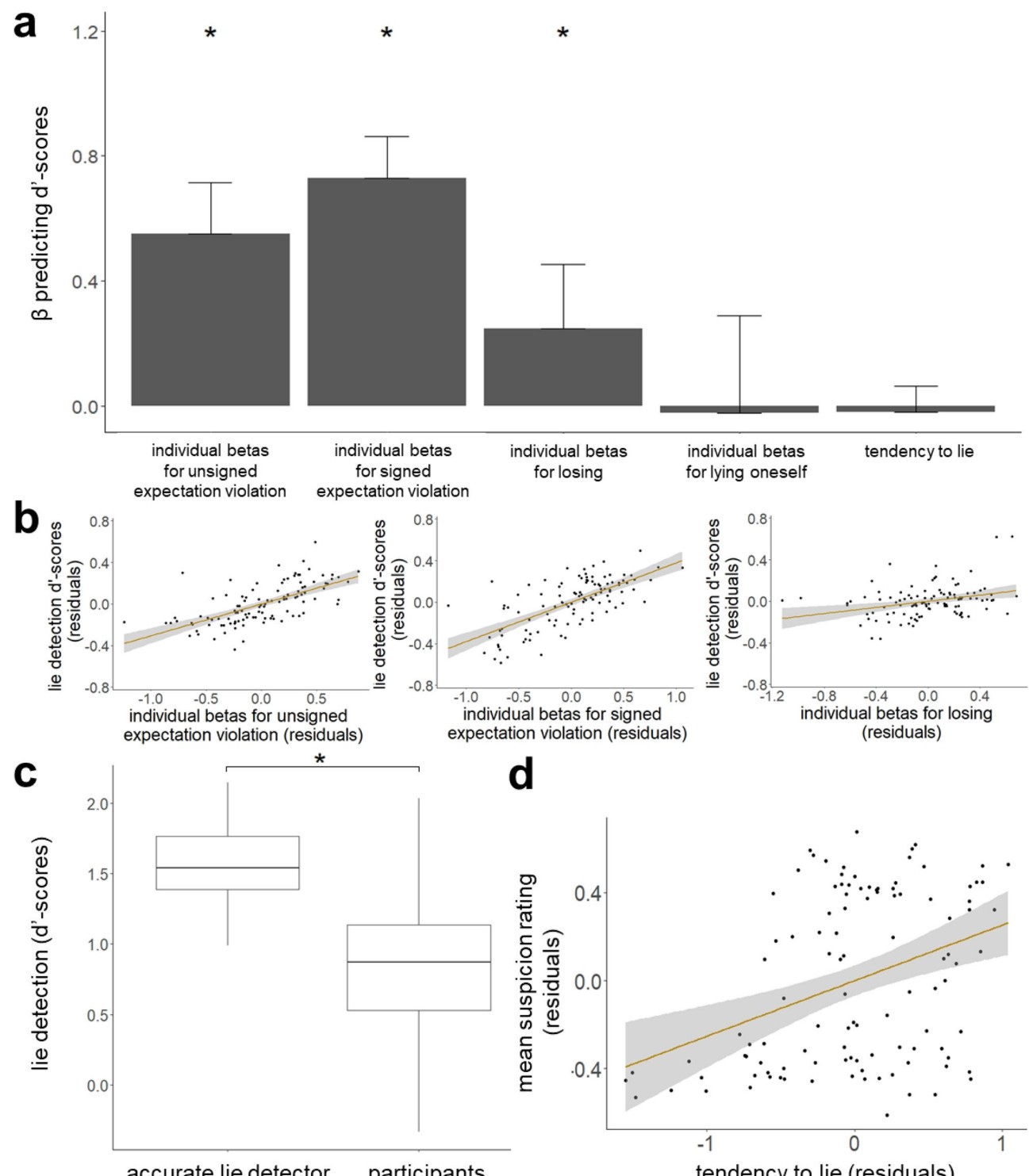

**Fig. 5 | Better lie detection is associated with reliance on statistical likelihoods and outcomes, but not with reliance on one's own lying behaviour. a** Better discernment (d'-scores) across individuals (N = 108) was related to greater weight assigned to greater beta coefficients of signed expectation violation (standardised β = 0.732, t(98) = 9.16, *p* < 0.001, 95%-CI = [0.466; 0.994]), unsigned expectation violation (standardised β = 0.551, t(98) = 8.5, *p* < 0.001, 95%-CI = [0.228; 0.875]) and losing (standardised β = 0.248, t(98) = 3.31, *p* = 0.001, 95%-CI = [−0.159; 0.655]), but not participants' betas for their own lying (standardised β = −0.023, t(98) = −0.338, *p* = 0.736, 95%-CI = [−0.64; 0.594]), nor their tendency to lie (standardised β = −0.02, t(98) = −0.31, *p* = 0.758, 95%-CI = [−0.186; 0.146]). * *p* < 0.001 **b** Partial regressions showing that across individuals, greater beta estimates that relate suspicion to unsigned and signed expectation violations and losing were associated with discernment (d'-

scores). Shaded areas indicate 95% confidence intervals. Dots represent individual participants. **c** An accurate lie detector's suspicion model that does not consider information on when participants lie themselves outperforms human lie detection (Wilcoxon signed rank test = 5595, *p* < 0.001, N = 108). Horizontal lines indicate median values, boxes indicate 25–75% interquartile range, whiskers indicate 1.5 × interquartile range. * *p* < 0.001 **d** People who lie more are more suspicious of others, consistent with Experiment 1. The relationship between tendency to lie (i.e. proportion of trials on which an individual lied) and mean suspicion rating is shown, controlling for age, gender, education level and CRT-scores (standardised β = 0.381, t(102) = 4.174, *p* < 0.001, 95%-CI = [0.099; 0.63]). Shaded area indicates 95% confidence interval. Dots represent individual participants.

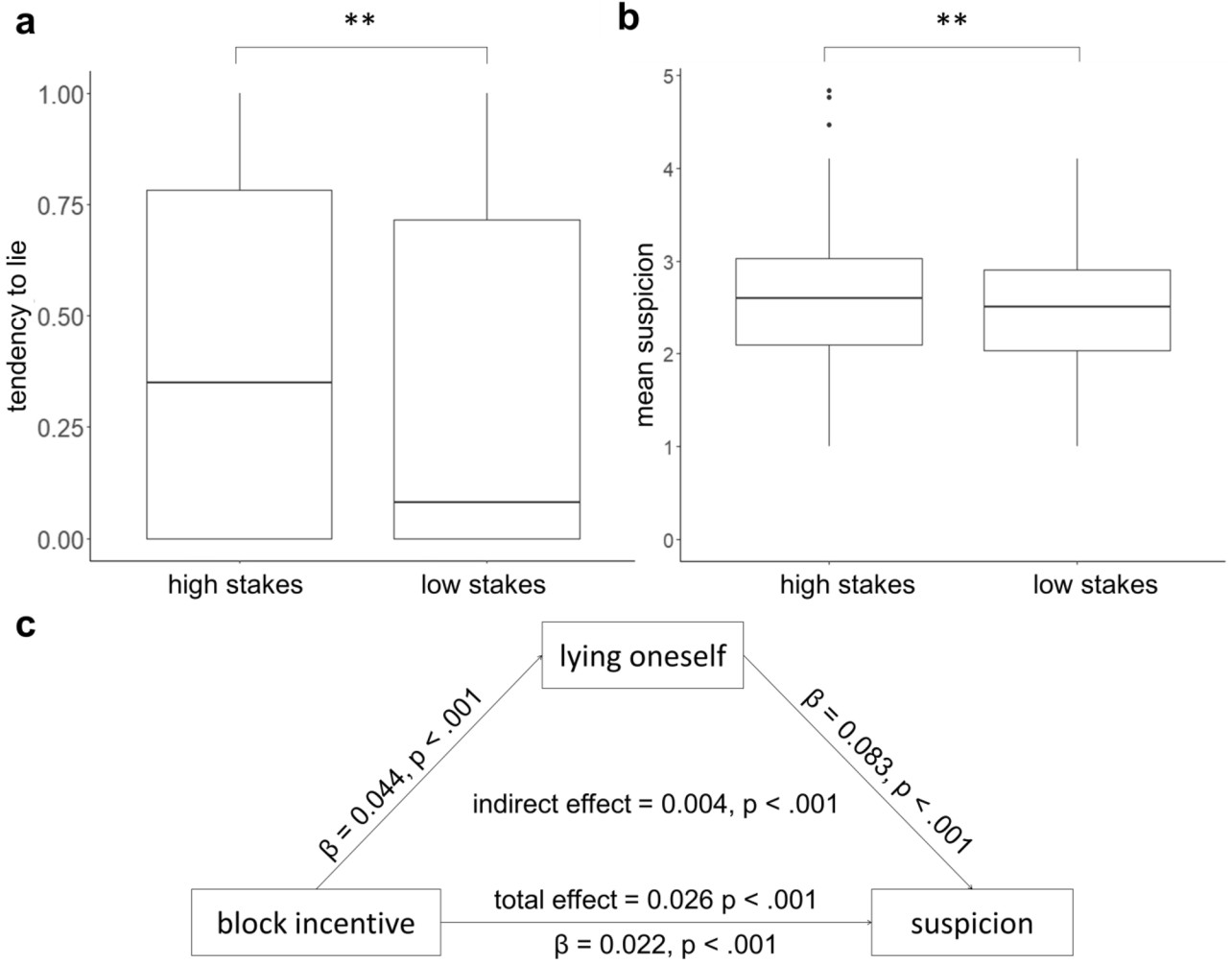

**Fig. 6 | People are more suspicious of others when they lie more themselves.**
**a** Participants (N = 100) lied more in the high stakes block than in the low stakes block (Wilcoxon signed rank test = 379.5, p = 0.002). **b** Participants (N = 100) were more suspicious of others when they themselves were confronted with higher stakes than lower stakes (Wilcoxon signed rank test = 1306, p = 0.002), despite the fact that the average frequency of other people's lies was constant. Dots represent individual participants. Horizontal lines indicate median values, boxes indicate 25–75%

interquartile range, whiskers indicate 1.5 × interquartile range. ** p < 0.01
**c** Mediation shows that suspicion was greater when block incentive was larger (total effect = 0.026, p < 0.001) and this effect was partially mediated by when participants lied (indirect effect = 0.004, p < 0.001). Participants lied more in the high stakes block (β = 0.044, p < 0.001) and suspicions increased when participants lied (β = 0.083, p < 0.001).

incentive (total effect = 0.026, p < 0.001, Fig. 6c) and that this effect was indeed partially mediated by when participants lied (indirect effect = 0.004, p < 0.001). Participants lied more in the high stakes block (β = 0.044, p < 0.001) and suspicions increased when participants lied (β = 0.083, p < 0.001). These results support the conclusion that when people lie more themselves, they assume others do the same.

We then set out to replicate the results of Experiment 1. To that end, we ran the same exact analysis as in Experiment 1 on all trials. With the model averaging procedure, we found that suspicion within participants was driven by the same four cues as in Experiments 1 and 2 (Fig. 7). Participants were more suspicious of others on trials where they lied themselves (mean weighted β = 0.079, 95%-CI = [0.06, 0.099]), when the other participant reported an unlikely card colour that benefited the other participant (i.e., signed expectation violation; mean weighted β = 0.31, 95%-CI = [0.234; 0.385]), when the other participant's card report was generally surprising (i.e., unsigned expectation violation; mean weighted β = 0.194, 95%-CI = [0.146; 0.241]) and when they lost (mean weighted β = 0.087, 95%-CI = [0.065; 0.108]). We confirm these findings using a leave-one-out cross-validation procedure and also after excluding participants (21%) who did not believe they were playing with other human participants (see Supplementary Tables 2a, 3). These results corroborate that people use their own

behaviour, statistical likelihood estimations and motivation-related cues (i.e., losing) to infer others' truthfulness.

Consistent with Experiment 1, we found that the suspicion of the accurate lie detector was associated with signed expectation violation (mean weighted β = 0.305, 95%-CI = [0.256; 0.353]) and general surprise (i.e., unsigned expectation violation; mean weighted β = 0.336, 95%-CI = [0.282; 0.389]), but not with when participants lied themselves (mean weighted β = 0, 95%-CI = [0; 0]). We also found an accurate lie detector's suspicion is weakly informed by when participants lost (mean weighted β = 0.025, 95%-CI = [0.019; 0.031]). The non-overlapping CIs of humans' and the accurate lie detector's betas imply that humans under-rely on unsigned expectation violation and over-rely on their own lying behaviour and losing (Fig. 7). These results align with the findings in Experiments 1 and 2.

Next, we examined what cues led to better discernment. Participants had an average d'-score of 0.77(SD = 0.579) and correctly identified 36% (M = 5.4, SD = 3.73) out of the circa 15 actual lies on average and wrongly suspected lies on 15% of the trials (M = 6.74; SD = 6.85). As in Experiment 1 and 2: better discernment (i.e., higher d'-scores) was associated with greater beta coefficients for unsigned expectation violation (standardised β = 0.581, t(90) = 7.94, p < 0.001, 95%-CI = [0.198; 0.965]), signed expectation violation (standardised β = 0.492, t(90) = 6.28, p < 0.001, 95%-CI = [0.193; 0.79])

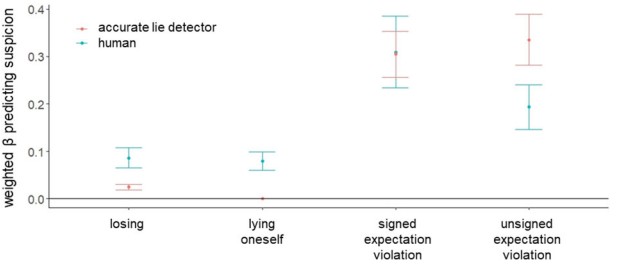

**Fig. 7 | Replication of cues driving human suspicion as compared to an accurate lie detector.** We constructed models based on every possible combination of one up to all four cues (signed and unsigned expectation violation, lying oneself, losing). and fitted these 15 models to participants' suspicion ratings (N = 100), using linear mixed-effects models with fixed and random intercepts and fixed and random slopes. The standardised beta coefficients were then weighted by the fitted model's BIC (Bayesian Information Criterion) and summed across all models that included the respective cue. This procedure revealed that all four cues were significantly associated with suspicion in humans (turquoise). The same procedure was applied to the suspicion of a simulated, accurate lie detector (that is, an agent which would always be suspicious when someone lies and not otherwise). Suspicion of this accurate lie detector was associated with signed and unsigned expectation violation and losing, but not with information about participants' own lying behaviour (red). Points represent weighted beta estimates, whiskers depict the upper and lower bounds of the 95% confidence intervals (CIs).

and losing (standardised β = 0.416, t(90) = 5.55, p < 0.001, 95%-CI = [−0.014; 0.847]), but not self-projecting one's own lying (standardised β = 0.008, t(90) = 0.11, p = 0.912, 95%-CI = [−0.756; 0.772]), nor participants' overall tendency to lie (standardised β = 0.113, t(90) = 1.47, p = 0.145, 95%-CI = [−0.114; 0.339]). We also found that those identifying as female had lower d'-scores (standardised β = −0.158, t(90) = −2.14, p = 0.035, 95%-CI = [−0.335; 0.02]) as in Experiment 2 (Fig. 8a, b). As in Experiments 1 and 2, the accurate detector outperforms humans (M = 1.7, SD = 0.226; Wilcoxon signed rank test = 4931, p < 0.001). See Fig. 8c. Thus, lie detection is consistently related to using statistical likelihood estimations and not one's own behaviour.

Lastly, consistent with Experiments 1 and 2, we show that participants' overall suspicion (i.e., mean suspicion ratings) was predicted by their tendency to lie (standardised β = 0.287, t(94) = 3.078, p = 0.003, 95%-CI = [−0.026; 0.6]) (see Fig. 8d). We also found that male participants were more suspicious of others (standardised β = −0.33, t(94) = 3.468, p < 0.001, 95%-CI = [−0.591; −0.069]), though this was not observed in Experiments 1 and 2.

Altogether, these results provide consistent evidence that reliance on statistical cues is associated with better discernment, but reliance on one's own tendency to lie is not.

## Discussion

Uncovering the cognitive processes that prevent people from detecting lies is crucial for developing tools to curb incidents of scams and fraud. Here, we reveal that poor lie detection is related to a suboptimality in the cues people use when assessing potential lies. In particular, over-reliance on one own's behaviour and outcomes and under-reliance on statistical cues.

We first find that in generating suspicion, participants used cues related to one's own behaviour (i.e., self-projection), theory of mind (i.e., inferred motivation of others) and statistical cues. As for the former, we find that participants use their own lying behaviour to infer the truthfulness of others' statements. On trials when a participant lied themselves, they were more suspicious of others. This result fits nicely with a body of literature in other domains that shows that people use their own behaviour to understand and predict the behaviour of others[27–34]. For example, people use their own mood and moral beliefs to understand and explain that of others[55–57]. In line with theory of mind reasoning, people also use cues related to other people's inferred motivation to lie[20]. As for statistical cues, we find that individuals

assess honesty by considering how likely the outcome conveyed by the other person is, given prior knowledge of the statistical likelihoods of such outcomes.

Across individuals, reliance on statistical likelihoods was associated with discernment accuracy, while reliance on one's own lying behaviour was not. In fact, participants used their own lying behaviour and outcomes to predict whether other people lied, despite this cue being uninformative, while under-using more predictive statistical cues. This was observed by comparing the weights participants assigned to different cues to that of a model trained on the ground truth. The beta coefficients of the model trained on the ground truth revealed significant positive weights assigned to statistical cues for predicting others' dishonesty and zero weight to the self-projection cue. Thus, the findings suggest that poor lie detection is partially due to humans' under-reliance on statistical cues, perhaps in favour of other, unhelpful cues.

Our findings imply that people may be less accurate in detecting lies in environments where true statistics are distorted. For example, on social media platforms, algorithms that recommend content similar to what people interacted with before[58] may inflate the apparent likelihood of certain information, including misinformation. This may lead to inaccurate expectations and subsequently reduce one's ability to reject falsehoods (due to distorted expectation violation signals).

The finding that subjects are more suspicious of others when they themselves lie, may indicate that people use their own behaviour to infer the behaviour of others ('If I lie, others probably lie too') or that people's perception of others alters their own behaviour ('Others are lying, so I will too'), or both. To test these possibilities, we conducted two additional experiments. In Experiment 2, we manipulated the percentage of times others lied and examined if participants lied more. While participants were more suspicious of others (because signed and unsigned expectation violation were high and participants lost more) they did not lie more themselves. Next, in Experiment 3 we manipulated participants' lying behaviour by altering their incentive to lie, without changing others' lying tendency. We found that when participants lied themselves due to high incentives, they were more suspicious of others, even though other people's rate of lying did not change. Together, these studies provide greater support for the idea that people use their own behaviour to infer the behaviour of others, rather than that to the idea that people's perception of others alters their own behaviour.

Our results are based on a task in which (i) the participant is both the sender and assessor of lies and in which we (ii) present participants with real statements from others. In contrast, most studies assign only one role to a participant – either an observer or sender of potential lies, not both[3,25], and/or do not provide participants with naturally occurring data, but with pre-fixed trials[3,20]. Moreover, unlike many other studies, we did not instruct people to lie, nor presented any information on others' lying frequency[23,59,60]. Indeed, roughly a third of the participants rarely lied and few lied more than half the time. This is in line with prior work[25,61–66].

## Limitations

Experiments 1 and 2 were performed at different times and thus conclusions based on their comparison are tentative. Note also that we are not suggesting that statistical cues are necessarily provided in numerical form in real life (online or offline), but rather that people have an internal access to such rough estimates. Similarly, people likely have a sense of their own likelihood to engage in a certain type of lie (e.g., one may have a sense that the likelihood they will present oneself as a wealthy heiress is zero, but the likelihood that one will say they liked a gift which they did not is ~90%) and use these estimates to infer other's likelihood of doing the same. Future studies can examine more such real-life instances.

## Conclusion

In summary, we show that people rely on a range of cues to infer other people's honesty, including statistical likelihoods and their own behaviour.

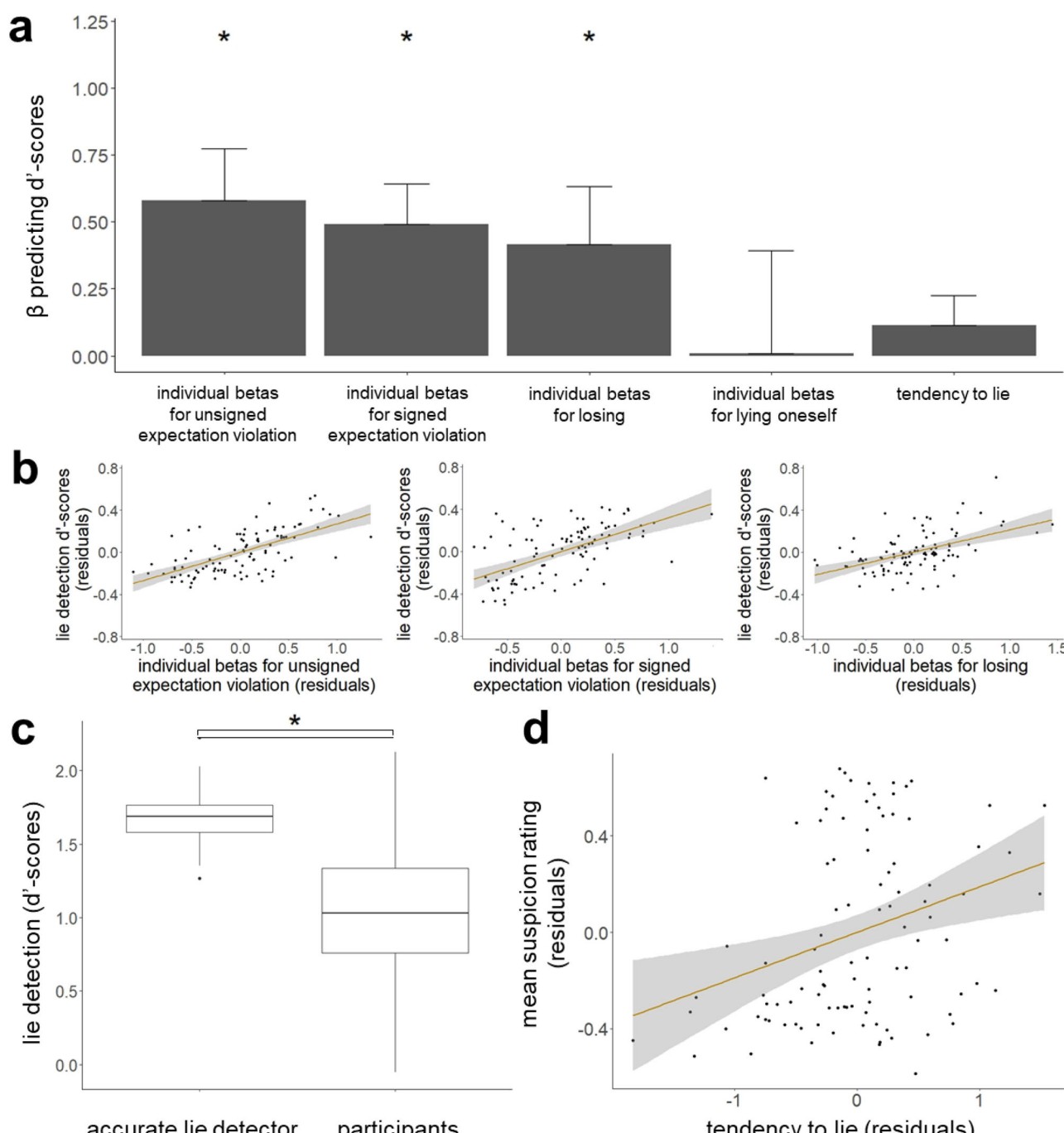

**Fig. 8 | Better lie detection is associated with reliance on statistical likelihoods and outcomes, but not with projecting one's own lying behaviour. a** Better discernment (d'-scores) across individuals (N = 100) was related to greater weight assigned to unsigned expectation violation (standardised β = 0.581, t(90) = 7.94, p < 0.001, 95%-CI = [0.198; 0.965]), signed expectation violation (standardised β = 0.492, t(90) = 6.28, p < 0.001, 95%-CI = [0.193; 0.79]) and losing (standardised β = 0.416, t(90) = 5.55, p < 0.001, 95%-CI = [−0.014; 0.847]), but not self-projecting one's own lying (standardised β = 0.008, t(90) = 0.11, p = 0.912, 95%-CI = [−0.756; 0.772]), nor participants' overall tendency to lie (standardised β = 0.113, t(90) = 1.47, p = 0.145, 95%-CI = [−0.114; 0.339]). Whiskers indicate standard error (SE). * p < 0.001 **b** Partial regressions showing that across individuals, greater beta estimates that relate suspicion to unsigned and signed expectation violation and losing were associated with discernment (d'-scores).

Shaded areas indicate 95% confidence intervals. Dots represent individual participants. **c** An accurate lie detector's suspicion model that does not consider information on when participants lie themselves outperforms human lie detection (Wilcoxon signed rank test = 4931, p < 0.001, N = 100). Horizontal lines indicate median values, boxes indicate 25–75% interquartile range, whiskers indicate 1.5 × interquartile range. * p < 0.001 **d** People who lie more are more suspicious of others, consistent with Experiment 1 and 2. The relationship between tendency to lie (i.e. proportion of trials on which an individual lied) and mean suspicion rating is shown, controlling for age, gender, education level and CRT-scores (standardised β = 0.287, t(94) = 3.08, p = 0.003, 95%-CI = [−0.026; 0.6]). Shaded area indicates 95% confidence interval. Dots represent individual participants.

However, while likelihood estimations are helpful in improving lie detection, reliance on one's own behaviour is not. To improve lie detection, providing contextual information may be helpful[19], as contextual cues can inform people's expectations about event likelihoods.

## Data availability

All raw and processed data used for the main analyses and supplementary information are freely accessible in .csv format via GitHub: https://github.com/affective-brain-lab/suspicion-, https://doi.org/10.5281/zenodo.10606288.

## Code availability

The custom code used to produce the results are freely accessible via GitHub: https://github.com/affective-brain-lab/suspicion-, https://doi.org/10.5281/zenodo.10606288. All analyses were carried out with R version 4.1.3.

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

## Acknowledgements

We thank Moshe Glickman, Bastien Blain, Valentina Vellani, Laura Globig, Irene Cogliati Dezza and India Pinhorn for their feedback on earlier versions of the manuscript. This research was funded by the Dawes Centre for Future Crime to S.Y.Z. and a Wellcome Trust Senior Research Fellowship (214268/Z/18/Z) to T.S. The funders had no role in study design, data collection and analysis, decision to publish or preparation of the manuscript.

## Author contributions

Sarah Ying Zheng conceptualised and developed the experiments, collected and analysed all data and co-wrote the manuscript. Liron Rozenkrantz developed the experiments and provided guidance and feedback on earlier versions of the manuscript. Tali Sharot conceptualised and developed the experiments, provided guidance, checked the analyses and co-wrote the manuscript.

## Competing interests

The authors declare no competing interests.
