## [Peer Review File · Communications Psychology]

24th Jul 23

Dear Tali,

Thank you for your patience during the peer-review process. Your manuscript titled "Why humans are bad at detecting lies" has now been seen by 3 reviewers, whose comments are appended below. You will see that they find your work of some potential interest. However, they have raised quite substantial concerns that must be addressed. In light of these comments, we cannot accept the manuscript for publication, but would be interested in considering a revised version that fully addresses these serious concerns.

We hope you will find the Reviewers' comments useful as you decide how to proceed. Should additional work allow you to address these criticisms, we would be happy to look at a substantially revised manuscript. If you choose to take up this option, please highlight all changes in the manuscript text file, and provide a detailed point-by-point reply to the reviewers.

Editorially, we consider two areas of improvement key: first, the reviewers list a range of concerns regarding additional analyses that will support the strength of your conclusions. We ask you to address all of the listed uncertainties, especially those regarding confounds and model specification, through comprehensive additional statistical analysis. Relatedly, we ask that you respond to the reviewer requests for more visual presentation of the basic behavioural data. You may include up to 10 display items in the manuscript.

Second, the referees highlight that the interpretation of the results and their implications is too far removed from the data. We ask you to revise the work towards a more toned-out presentation of the work.

I include below two links, one to a template and one to a checklist. As you revise your work, please ensure that it complies with our formatting and reporting requirements as detailed in the checklist. This includes having a section titled "Limitations" in the Discussion section.

If the revision process takes significantly longer than five months, we will be happy to reconsider your paper at a later date, provided it still presents a significant contribution to the literature at that stage.

Please use the following link to submit your revised manuscript, point-by-point response to the Reviewers' comments with a list of your changes to the manuscript text (which should be in a separate document to any cover letter) and any completed checklist:

[link redacted]

Please do not hesitate to contact me if you have any questions or would like to discuss the required revisions further. Thank you for the opportunity to review your work.

Best wishes,

Marike, on behalf of Jennifer Bellingtier

Jennifer Bellingtier, PhD
Senior Editor
Communications Psychology

Marike Schiffer, PhD
Chief Editor
Communications Psychology

EDITORIAL POLICIES AND FORMATTING

Editorial Policy: Policy requirements (Download the link to your computer as a PDF.)

Furthermore, please align your manuscript with our format requirements, which are summarized on the following checklist:

Communications Psychology formatting checklist

and also in our style and formatting guide Communications Psychology formatting guide .

* TRANSPARENT PEER REVIEW: Communications Psychology uses a transparent peer review system. This means that we publish the editorial decision letters including Reviewers' comments to the authors and the author rebuttal letters online as a supplementary peer review file. However, on author request, confidential information and data can be removed from the published reviewer reports and rebuttal letters prior to publication. If your manuscript has been previously reviewed at another journal, those Reviewers' comments would not form part of the published peer review file.

* CODE AVAILABILITY: All Communications Psychology manuscripts must include a section titled

"Code Availability" at the end of the methods section. In the event of publication, we require that the custom analysis code supporting your conclusions is made available in a publicly accessible repository; please choose a repository that provides a DOI for the code; the link to the repository and the DOI must be included in the Code Availability statement. Publication as Supplementary Information will not suffice. We ask you to prepare and upload code at this stage, to avoid delays later on in the process.

* DATA AVAILABILITY:

All Communications Psychology research manuscripts must include a section titled "Data Availability" at the end of the Methods section or main text (if no Methods). More information on this policy, is available at <http://www.nature.com/authors/policies/data/data-availability-statements-data-citations.pdf>.

At a minimum the Data availability statement must explain how the data can be obtained and whether there are any restrictions on data sharing. Communications Psychology strongly endorses open sharing of data. If you do make your data openly available, please include in the statement:

We recommend submitting the data to discipline-specific, community-recognized repositories, where possible and a list of recommended repositories is provided at <http://www.nature.com/sdata/policies/repositories>.

If a community resource is unavailable, data can be submitted to generalist repositories such as [figshare](http://www.figshare.com) or [Dryad Digital Repository](http://www.dryad.org). Please provide a unique identifier for the data (for example a DOI or a permanent URL) in the data availability statement, if possible. If the repository does not provide identifiers, we encourage authors to supply the search terms that will return the data. For data that have been obtained from publicly available sources, please provide a URL and the specific data product name in the data availability statement. Data with a DOI should be further cited in the methods reference section.

REVIEWER EXPERTISE:

Reviewer #1 Deception/(dis)honesty, decision making, computational modelling

Reviewer #2 Deception/(dis)honesty, decision making, computational modelling

Reviewer #3 Deception/(dis)honesty, decision making, computational modelling

Reviewer #1 (Remarks to the Author):

Review of Zheng et al for Nature Communications Psychology

Summary: The authors present three experiments investigating the factors underlying people's

ability to detect lies. The experiments use a computerized card game task in which participants are first shown a random selection of blue and red cards (7 in total), they are then shown which card the computer randomly chooses for that trial – indicated via a yellow highlight box, they are then asked to report whether the selected card was “blue” or “red”. Finally, they are shown the card they reported alongside the card reported by another participant, whose response was randomly selected from those made by previous participants who had seen the same initial set of cards. If the card selected by the current player is the same as the one chosen by the randomly selected participant, then they tie, if the current player says “blue” and the other says “red” the current player wins, and if the previous participant says “blue” and the other says “red”, the previous participant wins. At the end of each trial the current player rates the honesty of the other player on this particular trial. The key findings are that when judging the honesty of the other player, participants tend to (over) rely on simple heuristics (assumptions about others’ tendency to lie based on their own behaviour) rather than statistical cues provided by the ratio of blue and red cards in the initial sample. The results are interpreted in terms of potential methods to combat online scams and phishing techniques.

Evaluation: This is a timely and well-written paper that presents an interesting and novel task for assessing people’s capability for detecting lies. The application to phishing scams, etc., is debatable and despite the ‘special pleading’ in the general discussion regarding the generalizability to real world situations I was not convinced. This, however, does not necessarily undermine the contribution, it simply requires toning down some of the conclusions and framing. I will return to this issue at the end of my review, first I will outline some of the aspects I think would need improving if a revision were offered by the Editor.

1. The task is a relatively complex one to communicate to participants in an online marketplace setting like Prolific. The various reputational concerns and demand characteristics inherent in online participation make this kind of experiment potentially difficult to interpret straightforwardly. The design involves not only the theory of mind element highlighted in the introduction (between the participant and the randomly selected other player on each trial) but also the ‘theory of the experimenter’s mind’ element. That is, the assumptions about what it is the experimenter is expecting the participant to do and the potential consequences for behaving in particular ways. For example, although the authors point out that the task does not require or necessarily invite lying, it does ask on every trial about the perceived honesty of the ‘other player’ thus, presumably, highlighting for the participant that this is a characteristic of interest. I would like more reassurance – perhaps from more details of the manipulation/instruction/attention/comprehension checks that participants were interpreting the task in the intended way and that they truly understood the stochastic nature of the card selections, and the elements of each trial that were the same (i.e., initial card array) and different (selected card) for them and the random other player. It would also be good to know if there was any debriefing or questioning regarding participants’ beliefs about the veracity of the experiment/experimenter (e.g., did they believe the selection data actually came from previous players?).

2. Following on from point 1, it was unclear to me where the data for the randomly selected previous participants came from for the participants in Experiment 1. I might have missed this information – were there previous pilot experiments (not reported?) that generated the responses that were selected from for the in the initial experiment? Was the same pool of past participants’ data used in all three experiments or was the pool added to as the experiments were completed? In Experiment 3 had the past participants faced a situation with the same payment contingencies as

used in Experiment 3, or ones similar to Expt 1 and 2?

3. I felt rather distant from the raw data. At the start of the results section what I wanted to see was how often, and how many people actually lied. Instead, we are presented straight away with regression models aimed at uncovering the factors predicting the ability to detect lying in others. Eventually, via mentions in the Figure captions and directions to the histograms in the Supplementary material we get to see these patterns, but I think this information should be given first. It is important to know that the base rate of lying in the task is low, with the modal response being not to lie at all. I don't think this is a big issue (although the skewed – non-normal - distribution has implications for the statistics examining tendency to lie – e.g., the t-tests in Experiment 3 comparing high and low stakes are surely inappropriate) – but it is important for interpreting the regressions.

4. Following on from point 3, in Experiment 2 can you report how many trials participants lied on, on average? The moderate shift in the proportion of people lying at least half the time is reported, but not the mean. From the histograms, it would appear that this does not change very much thus raising some doubt about the strength of the manipulation in increasing the rate of lying.

5. The demarcation between the 'heuristic' and 'statistical' elements in the regression models is a nice feature and supports the general take home message of the paper. However, I was concerned about the high correlation between the two measures of expectation violation. Given their formulation, this correlation is to be expected but it left me unsure about how/whether to think of these as independent contributors to either a participant's or the optimal model's ability to predict suspicion ratings. I was also not sure how to think about the large CIs around the coefficient estimates for the violations measures in the optimal model compared to the much smaller CIs for the participants (note that in the Figures the bars should be labelled as CIs in the caption). Can the authors say more about these features of the data?

6. Returning to the theme of how much these results tell us about why people fall for online scams, I was unconvinced by the idea that small vs. big lie detection is linked to people's tendency to engage in the former but not the latter. Is this not also partially a base-rate phenomenon (i.e., statistical inference) – we know that small (white) lies are far more common (whether we engage in them or not) and big fraud is much less common. If you had rigged the statistical structures in your experiment so that really big high stakes lies were very infrequent then wouldn't your optimal model have a harder time predicting suspicious behaviour? If you really think these results can "inform the design of digital interfaces" (the Abstract) then I think you need to expand on how in the paper itself.

Reviewer #2 (Remarks to the Author):

The manuscript deals with an important question. I appreciate the creative paradigm to study lying and suspicion. I also think the rationale of the three experiments are well justified, and the results are mostly straightforward. My points of criticism concern some of the framing in the manuscript, the (lacking) analysis and the relatively far-reaching conclusions the authors draw.

Predictors / cues that inform the suspicion judgment.

The authors conclude: "In particular, overweighting simple heuristics (namely an assumption that

others lie when we do) and underweighting computationally demanding assessments of statistical cues." It is not clear to me in what sense -- and the authors do not spell it out -- the cue or the predictor that others lie when we do or when I loose there other lied represent heuristics, while the statistical cues are not heuristics. What exactly is the conceptual differences between these predictors and on which basis can one characterize one set as simple heuristics and other as statistical cues.

Why not call all of them 'cues' of different complexity, and with as Brunswik would have said, different ecological validities. Indeed, a lens model analysis in which the ecological validities of the cues and the cue utilization validities of the cue are calculated would have build on past work and permit for insight and past benchmarks (see the various highly relevant papers on lens model analysis by Ken Hammond and colleagues; see also Hartwig, M., & Bond Jr, C. F. (2011). Why do lie-catchers fail? A lens model meta-analysis of human lie judgments. *Psychological bulletin*, 137(4), 643.).

Signal-detection analysis and cross-validation

I would have appreciated more information about the outcome of the SDT-analysis, especially about d' and how good or bad people are at the discernment judgments and also interindividual variation on the judgment criterion β . For instance, do people, who are more inclined to lie, have a different β than other participants. More generally, I would have found the SDT analysis, in combination with the a lens-model analysis, more intuitive, relative to the statistical parameters reported in the manuscript.

I would also suggest that the authors consider cross-validation procedures to determine the predictive accuracy of the various cues!

How general are the results?

On p. 22, the authors conclude: "while some may claim that studying suspicion in a controlled task cannot provide insights about 'real-world' behaviour, we beg to differ." I am not yet convinced by the authors' arguments concerning the utility of a controlled task. The issue is not whether the task is controlled or less controlled and involving more real-world aspects.

The key question is to what extent the structural features of the task, including the statistical cues, also exit in the real world, and whether we know anything about the validity of 'statistical cues' and what one could call the self-projection cue in the real world. In other world, only if we are confident that these structural features are representative for the probabilistic texture of the world outside of the laboratory (or at least parts of it), should we draw strong conclusions such as "these findings suggest that poor lie detection is partially due to overreliance on simple heuristics, namely an assumption that others behave as we do, and under-reliance on more costly computations" (p. 22).

Rationality of the 'lying' behavior

Finally, I propose the authors point out that their experiments implemented an incentive structure in which participant can maximize income by always reporting the color 'blue'. This meant that in cases where the actual color was 'red', the income-maximizing response is to always say 'blue'.

The question then is: do people infer that the experimenter has implemented an environment in which lying is the 'rational' thing to do? The experimenters also probed 'honesty' judgments. It thus seems that the experimenters' conveyed two conflicting messages as to what is the expected /

desired behavior in the study. Can the authors say a bit more about what kind of assumptions / expectations individual respondents formed about the study, and how those may have informed their behaviors.

Reviewer #3 (Remarks to the Author):

The manuscript presents an investigation of the heuristics that people use to detect lies. In three experiments, the authors compared participants' heuristics to an optimal lie detector and found that participants assessed others' honesty based on their own honesty, whereas the optimal lie detector did not. In addition, they found that being more suspicious of others' lying leads to more lies of one's own, and people who lie more themselves are more suspicious of others' honesty.

The work presents an interesting and relevant research topic in the field and makes a valuable contribution to the literature. However, I currently have some concerns regarding the task design and data analyses, which I elaborate on below.

1. The authors compared participants' heuristics to a statistical model trained using the actual feedback data (whether the player of interest actually lied or not). The statistical model relied on four features: the other player's behavior (lied or not), signed expectation violation, unsigned expectation violation, and who lost in the trial.

1a. The authors may have taken it too far by labeling this statistical model as an optimal lie detector, which suggests that no other model or strategy can surpass it in terms of lie detection. To establish its optimality, they need to demonstrate that this model outperforms all other reasonable candidate models, such as those incorporating additional features or employing non-linear combinations of the features. Failing to provide such evidence may leave some readers skeptical about its optimality.

1b. The signed expectation violation is computed based on one's motivation to lie (positive if the motivation is to lie and negative otherwise), and the unsigned expectation violation. In fact, if I understand it correctly, it's lying motivation multiplied by the unsigned expectation violation, or in other words, the interaction between the two. I wonder if the authors have explored the possibility of including lying motivation as a dummy variable in their analyses and how that might affect the results. I am suggesting this because, typically, when a two-way interaction is included in a regression model, it is common to include both main effects as well.

1c. How well did the optimal lie detector perform? I suggest that the authors report the accuracy and d'-scores of the detector and show us how much better the detector was compared to human judges. This will help readers evaluate the detector's performance.

1d. One of the main conclusions is that the optimal lie detector does not take into account the opponent's behavior when assessing whether the player has lied. This conclusion appears evident and does not necessarily require statistical analyses. Given that the behaviors of both players were

independent when conditioned on the shared presented card sets, it becomes apparent that, once controlling for the features of the card set that are related to lying probability (e.g., expectation violation and lying motivation), the opponent's behavior cannot be used to predict the other side's lying probability.

1e. Related to 1d, I suggest that the authors perform a more thorough comparison between the participants' heuristics and the optimal lie detector. For example, did the participants rely on the other features (e.g., signed/unsigned expectation violation) more or less than the optimal lie detector?

1f. Although human judges relied on their own behaviors to assess whether the other side lied, the strength of this tendency appeared to be relatively weak, especially when compared to the influence of signed expectation violation and unsigned expectation violation (see Figures 2A, 5A, and 9A). Considering the weak relationship observed in the aggregate-level data, exploring the relationship at the individual level would be useful. Specifically, it would be useful to determine the proportion of participants who showed a 'significant' reliance on their own behaviors when judging others' honesty. If this proportion is small, the authors may need to modify their conclusions regarding the extent of humans' over-reliance on their own behaviors when judging others. Moreover, it remains unclear from the current discussion whether the aggregate-level results were primarily driven by individual differences (i.e., individuals who lied more also believed others lied more) or from trial-by-trial variations (i.e., people perceiving others as more dishonest if they themselves lied in a specific trial). Further clarification on this point would be useful.

2. In experiment 1, in ~25% of the trials the participants observed the others lie, and in experiment 2, the rate was 50%. The authors found that participants lied more in experiment 2 than in experiment 1 and concluded that being more suspicious of others lying led to more lies of one's own. I'm a bit concerned by this conclusion because experiment 2 and experiment 1 were performed at different times, and thus several uncontrolled factors could have contributed to the observed difference in lying tendencies. To obtain more conclusive results, we need a single experiment, in which the percentage of trials where participants observe lying is manipulated within or between participants. I suggest the authors discuss this limitation, and state that their current conclusion is explorative.

3. Minor comments

3a. On page 10, the authors wrote, "association between losing and suspicion was related to discernment (i.e., to d' -scores), due to the fact that when the other participant lies, they do so in their favor, which thus increases the likelihood that the participant will lose" to explain the positive association between betas for losing and lie detection d' -scores among participants. Could the authors elaborate on this point more? If this heuristic (i.e., losing or not) helps participants improve judgment accuracy, why does the optimal lie detector ignore this factor?

3b. If possible, please include error bars for Figure 3a (and similar bar plots in the paper).

3c. Have the authors tested how betas for different factors relate to prediction accuracy (rather than d' -scores)? Do all the conclusions stay the same?

3d. In experiment 2, how were the trials selected to fix "the proportion of trials with lies at 50%"?

3e. On page 15 and page 19, the authors discussed the effect of gender, e.g., 'We also found an effect of gender on d' -scores (standardized $\beta = -0.147$, $t(98) = -2.325$, $p = .022$ '. I suggest the authors also comment on the direction of this result. Did male participants or female participants have higher d' -scores?

3f. In experiment 3, participants had different stakes in different blocks. Were they informed that the stakes for their opponents remained the same in different blocks? Otherwise, it seems sensible that some participants may assume that the other side would have the same stake as themselves and would base their honesty assessment on that.

3g. On page 21, the authors wrote, "The key result is that greater reliance on one's own behaviour (heuristic) led to more judgement errors, while reliance on statistical likelihoods and others' motivation led to better discernment." However, based on their analyses, there was no relationship between individual betas for lying oneself and lie detection d' -prime score. Therefore, reliance on one's own behavior (heuristic) was not associated with (instead of positively correlated with) the number of judgment errors one made.

3h. What are the catch trials (page 23)? Please provide descriptions of how those trials were generated.

We would like to thank the reviewers for a thorough and thoughtful commentary on our work. To ensure we address all the reviewers' comments and for ease of reference, we include the reviews below (in bold) followed by our response to each concern.

Reviewer #1

Summary: The authors present three experiments investigating the factors underlying people's ability to detect lies. The experiments use a computerized card game task in which participants are first shown a random selection of blue and red cards (7 in total), they are then shown which card the computer randomly chooses for that trial – indicated via a yellow highlight box, they are then asked to report whether the selected card was “blue” or “red”. Finally, they are shown the card they reported alongside the card reported by another participant, whose response was randomly selected from those made by previous participants who had seen the same initial set of cards. If the card selected by the current player is the same as the one chosen by the randomly selected participant, then they tie, if the current player says “blue” and the other says “red” the current player wins, and if the previous participant says “blue” and the other says “red”, the previous participant wins. At the end of each trial the current player rates the honesty of the other player on this particular trial. The key findings are that when judging the honesty of the other player, participants tend to (over) rely on simple heuristics (assumptions about others' tendency to lie based on their own behaviour) rather than statistical cues provided by the ratio of blue and red cards in the initial sample. The results are interpreted in terms of potential methods to combat online scams and phishing techniques.

Evaluation: This is a timely and well-written paper that presents an interesting and novel task for assessing people's capability for detecting lies. The application to phishing scams, etc., is debatable and despite the ‘special pleading’ in the general discussion regarding the generalizability to real world situations I was not convinced. This, however, does not necessarily undermine the contribution, it simply requires toning down some of the conclusions and framing. I will return to this issue at the end of my review, first I will outline some of the aspects I think would need improving if a revision were offered by the Editor.

- Thank you for the positive evaluation. As detailed below, we have toned down our speculation regarding generalizability to online platforms and address all other improvements suggested by the reviewer below.

REQUEST FOR MORE DATA ABOUT COMPREHENSION AND DEBRIEF

The task is a relatively complex one to communicate to participants in an online marketplace setting like Prolific. The various reputational concerns and demand characteristics inherent in online participation make this kind of experiment potentially difficult to interpret straightforwardly. The design involves not only the theory of mind element highlighted in the introduction (between the participant and

the randomly selected other player on each trial) but also the ‘theory of the experimenter’s mind’ element. That is, the assumptions about what it is the experimenter is expecting the participant to do and the potential consequences for behaving in particular ways. For example, although the authors point out that the task does not require or necessarily invite lying, it does ask on every trial about the perceived honesty of the ‘other player’ thus, presumably, highlighting for the participant that this is a characteristic of interest. I would like more reassurance – perhaps from more details of the manipulation/ instruction/ attention/ comprehension checks that participants were interpreting the task in the intended way and that they truly understood the stochastic nature of the card selections, and the elements of each trial that were the same (i.e., initial card array) and different (selected card) for them and the random other player.

- We thank the reviewer for prompting us to detail our comprehension checks, attention checks and debriefing questionnaire. Indeed, we were very careful to verify participants’ attention and comprehension of the task. To ensure that participants fully understood the cards task, we utilized seven comprehension check questions after the task instructions. These are now added to the **Supplementary Materials** (pages 3-4) and referenced in the Results. These questions include an item to test if participants understood that the initial card array was the same for the other person and the participant (item 1), the stochastic nature of the game (item 2), as well as four items on all possible trial scenarios (items 3-6) and that participants understood that they saw past participants’ responses in the task (item 7). Participants could attempt each question twice. If they failed twice at one question, participants could not continue the study. In other words, all participants who moved on to the task were successful at the comprehension checks. A total of 29 individuals (8.6%) could not move on to the task due to too many failed comprehension check question attempts.
- There were three attention checks in each block of the task, i.e., nine in total in Experiments 1 and 2, and six in Experiment 3. On attention check trials participants were asked to select a specific option on the honesty rating scale and submit the response (e.g., ‘Please select the third option from the left’). We added these details to the Methods under *Task procedure* (page 27). Three participants failed more than 33% of the attention check trials in Experiment 1 and one participant in Experiment 2, and were therefore excluded from analyses. In Experiment 3, all participants passed all attention check trials. The exclusion criterion and number of participants who were excluded subsequently are described in the Methods (page 25-26).

It would also be good to know if there was any debriefing or questioning regarding participants’ beliefs about the veracity of the experiment/experimenter (e.g., did they believe the selection data actually came from previous players?).

- Indeed, we included a debriefing questionnaire that probed participants’ beliefs about the veracity of the experiment. There were nine post-task questions in total which we now detail in the **Supplementary Materials** (pages 4-9). To examine

whether participants believed the data actually came from previous participants, we asked them to indicate how many actual past participants they thought they observed. Overall, the vast majority of participants believed they saw past participants' responses. Only seven out of 102 participants in Experiment 1, 17 out of 108 participants in Experiment 2 and 21 out of 100 participants in Experiment 3 responded "zero". That is, they thought the other persons' responses did not come from previous players. We ran additional analyses on the data for each experiment after excluding these participants and found that such exclusion did not change the results. We report these new findings in the Supplementary Materials (page 11).

REQUEST FOR MORE DETAILS ABOUT STUDY

Following on from point 1, it was unclear to me where the data for the randomly selected previous participants came from for the participants in Experiment 1. I might have missed this information – were there previous pilot experiments (not reported?) that generated the responses that were selected from for the in the initial experiment? Was the same pool of past participants' data used in all three experiments or was the pool added to as the experiments were completed?

- For the first twenty participants in Experiment 1, we sampled responses from pilot studies that used the same task with either 60 or 90 trials in total (pooled $N_{\text{participants}}=50$, pooled $N_{\text{trials}}=4200$; see page 26). After this, we continuously added participants' responses to the response pool to sample from for every next batch of up to twenty participants throughout Experiments 1 and 2 (see page 26). For Experiment 3, we used the same trials that participants saw in Experiment 1 (see page 27).

In Experiment 3 had the past participants faced a situation with the same payment contingencies as used in Experiment 3, or ones similar to Expt 1 and 2?

The past participants of Experiment 3 were the participants from Experiment 1 and 2, who faced different contingencies to the "live" participants in Experiment 3. This was because the purpose of Experiment 3 was to examine whether higher stakes would lead participants to lie more and become more suspicious of others, even though they were observing lie rates that were the same as Experiment 1 (see page 17).

ORDER OF DATA PRESENTATION, USE NON PARAMTRIC TESTS

I felt rather distant from the raw data. At the start of the results section what I wanted to see was how often, and how many people actually lied. Instead, we are presented straight away with regression models aimed at uncovering the factors predicting the ability to detect lying in others. Eventually, via mentions in the Figure captions and directions to the histograms in the Supplementary material we get to see these patterns, but I think this information should be given first. It is important

to know that the base rate of lying in the task is low, with the modal response being not to lie at all.

- Following the reviewer's comment, we now alter the order of result presentation to start with a description of participants' behaviour first and only then the modelling results (page 6).

I don't think this is a big issue (although the skewed – non-normal - distribution has implications for the statistics examining tendency to lie – e.g., the t-tests in Experiment 3 comparing high and low stakes are surely inappropriate) – but it is important for interpreting the regressions.

- Following the reviewer's comment, we now replaced the t-tests with Wilcoxon signed rank sum tests. All results remain the same.

MORE DETAILS REGARDING LYING FREQUENCY

Following on from point 3, in Experiment 2 can you report how many trials participants lied on, on average? The moderate shift in the proportion of people lying at least half the time is reported, but not the mean. From the histograms, it would appear that this does not change very much thus raising some doubt about the strength of the manipulation in increasing the rate of lying.

- This is a very good point. As a reminder, Experiment 1 showed that participants who lied more were more suspicious of others. This finding may indicate that people use their own behaviour to infer the behaviour of others ('If I lie, others probably lie too') or that people's perception of others alters their own behaviour ('Others are lying, so I will too'), or both. To test the second option ('Others are lying, so I will too'), we had participants observe more lies in Experiment 2. They were indeed more suspicious in Experiment 2 than in Experiment 1 (Mann-Whitney $U = 7116$, $p < .001$), due to the fact that both signed and unsigned expectation violations were larger in Experiment 2 compared to Experiment 1 (signed expectation violation: Mann-Whitney $U = 10092$, $p < .001$; unsigned expectation violation: Mann-Whitney $U = 10875.5$, $p < .001$) and participants in Experiment 2 lost more than in Experiment 1 (Mann-Whitney $U = 6628.5$, $p = .011$), because others lied more than they did. Yet, participants' mean lying did not increase in Experiment 2 (28.3% of the trials) compared to Experiment 1 (23.3% of the trials; Mann-Whitney $U = 5730.5$, $p = .611$). This suggests that the result of Experiment 1 that 'participants who lied more were more suspicious of others', which we replicate in Experiment 2, is unlikely to be explained as 'people's perception of others alters their own behaviour'. We updated the conclusions throughout the manuscript accordingly (page 13-14, 17 and 23) The manuscript takeaways are now much clearer and we thank the reviewer for that.

CORRELATION BETWEEN MEASURES, AND CI

The demarcation between the ‘heuristic’ and ‘statistical’ elements in the regression models is a nice feature and supports the general take home message of the paper. However, I was concerned about the high correlation between the two measures of expectation violation. Given their formulation, this correlation is to be expected but it left me unsure about how/whether to think of these as independent contributors to either a participant’s or the optimal model’s ability to predict suspicion ratings.

- The correlations between signed and unsigned expectation violation is 0.558 in Experiment 1, 0.638 in Experiment 2 and 0.556 in Experiment 3. These are of only moderate size (see Supplementary Materials, pages 9 and 10), thus they can both be included in a model to quantify independent contributions. Signed expectation violation (also known as signed prediction errors, or signed surprise) is theoretically distinct from unsigned expectation violation (also known as unsigned prediction errors, or unsigned surprise). The former indicates how good the surprise is, while the latter only indicates the magnitude of the surprise. We and others have used these two measures to probe behaviour and neural correlates before (e.g., Sharot et al., 2011), and they have distinct neural signatures.

I was also not sure how to think about the large CIs around the coefficient estimates for the violations measures in the optimal model compared to the much smaller CIs for the participants (note that in the Figures the bars should be labelled as CIs in the caption). Can the authors say more about these features of the data?

- We thank the reviewer for noticing this. This was an error, which we have now corrected (see pages 9, 14, 19). The CIs of participants and accurate lie detector models are in fact of comparable size.

TONE DOWN

Returning to the theme of how much these results tell us about why people fall for online scams, I was unconvinced by the idea that small vs. big lie detection is linked to people’s tendency to engage in the former but not the latter. Is this not also partially a base-rate phenomenon (i.e., statistical inference) – we know that small (white) lies are far more common (whether we engage in them or not) and big fraud is much less common. If you had rigged the statistical structures in your experiment so that really big high stakes lies were very infrequent then wouldn’t your optimal model have a harder time predicting suspicious behaviour? If you really think these results can “inform the design of digital interfaces” (the Abstract) then I think you need to expand on how in the paper itself.

In the discussion we had speculated regarding the implication of our results regarding the likelihood of detecting small vs. big lies. We take the reviewer’s comment that this aspect was not tested in the experiment and other factors may be at play. Thus, we deleted this sentence.

We do believe the results can inform the design of digital interfaces. In particular, the results suggest that providing people with information on the likelihood of relevant population statistics is an especially promising way to help people make veracity judgements (page 2 and 23). For instance, AI can be used to infer from text (e.g., an email, a Tweet) a claim made and provide the user with relevant statistics regarding that claim (Abstract, Discussion).

Reviewer #2:

The manuscript deals with an important question. I appreciate the creative paradigm to study lying and suspicion. I also think the rationale of the three experiments are well justified, and the results are mostly straightforward. My points of criticism concern some of the framing in the manuscript, the (lacking) analysis and the relatively far-reaching conclusions the authors draw.

- We thank the reviewer for this positive assessment and address all comments below.

Predictors / cues that inform the suspicion judgment.

The authors conclude: "In particular, overweighting simple heuristics (namely an assumption that others lie when we do) and underweighting computationally demanding assessments of statistical cues." It is not clear to me in what sense -- and the authors do not spell it out -- the cue or the predictor that others lie when we do or when I loose there other lied represent heuristics, while the statistical cues are not heuristics. What exactly is the conceptual differences between these predictors and on which basis can one characterize one set as simple heuristics and other as statistical cues. Why not call all of them 'cues' of different complexity, and with as Brunswik would have said, different ecological validities.

- We thank the reviewer for the great suggestion to use the term "cues" instead. We revised the manuscript accordingly and describe them as related to self-projection, or statistical likelihood to better demarcate their conceptual difference.

Indeed, a lens model analysis in which the ecological validities of the cues and the cue utilization validities of the cue are calculated would have build on past work and permit for insight and past benchmarks (see the various highly relevant papers on lens model analysis by Ken Hammond and colleagues; see also Hartwig, M., & Bond Jr, C. F. (2011). Why do lie-catchers fail? A lens model meta-analysis of human lie judgments. Psychological bulletin, 137(4), 643.).

- We thank the reviewer for pointing us to Brunswik's lens model. We now describe the "accurate lie detector model" as the "ecological criterion (Y_e)" and participants' suspicion as "human judgments (Y_s)" as in the adapted Brunswik model for studying human judgement and decision-making by Cooksey (1996), and added the ecological and cue utilization validities according to this framework

in the **Supplementary Materials** (pages 13-15). Our main analysis can in fact be regarded as a statistically more intricate variation on Brunswik's framework that allows us to control for multiple factors at the same time.

Signal-detection analysis and cross-validation

I would have appreciated more information about the outcome of the SDT-analysis, especially about d' and how good or bad people are at the discernment judgments and also interindividual variation on the judgment criterion beta. For instance, do people, who are more inclined to lie, have a different beta than other participants. More generally, I would have found the SDT analysis, in combination with the lens-model analysis, more intuitive, relative to the statistical parameters reported in the manuscript.

- We thank the reviewer for prompting us to add information about the SDT analysis. We found that participants had an average d' -score of 0.937 (SD = 0.534) in Experiment 1, 0.833 (SD = 0.482) in Experiment 2 and 0.77 (SD = 0.579) in Experiment 3. Participants correctly identified less than half of the lies on average across the experiments (Experiment 1: 44.2%; Experiment 2: 47.1%; Experiment 3: 36%). They wrongly suspected lies on up to one fifth of the trials on average (Experiment 1: 15.1%; Experiment 2: 20.9.5%; Experiment 3: 15%). We now report these in the revised manuscript (page 10, 15 and 20).
- Following the reviewer's suggestion, we also ran a regression predicting participants' beta criterions from their betas related to the suspicion cues, demographics, questionnaire scores and their tendency to lie. Higher beta criterions were related to higher betas relating suspicion to unsigned expectation violation (Experiment 2: standardised $\beta = 0.316$, $t(98) = 3.39$, $p = .001$; Experiment 3: standardised $\beta = 0.363$, $t(90) = 3.71$, $p < .001$; Experiment 1: standardised $\beta = 0.182$, $t(89) = 1.74$, trend $p = .085$). No other predictors were consistently significant across experiments. We include these results in the Supplementary Materials (page 13).

I would also suggest that the authors consider cross-validation procedures to determine the predictive accuracy of the various cues!

- Yes, we agree. We now provide leave-one-out cross-validation analyses to test the consistency of the suspicion predictors in the **Supplementary Materials** (pages 10 and 11). We find that the key results remain the same.

How generalisable are the results?

On p. 22, the authors conclude: "while some may claim that studying suspicion in a controlled task cannot provide insights about 'real-world' behaviour, we beg to differ." I am not yet convinced by the authors' arguments concerning the utility of a controlled task. The issue is not whether the task is controlled or less controlled and involving more real-world aspects. The key question is to what extent the structural features of the task, including the statistical cues, also exist in the real

world, and whether we know anything about the validity of 'statistical cues' and what one could call the self-projection cue in the real world. In other words, only if we are confident that these structural features are representative for the probabilistic texture of the world outside of the laboratory (or at least parts of it), should we draw strong conclusions such as "these findings suggest that poor lie detection is partially due to overreliance on simple heuristics, namely an assumption that others behave as we do, and under-reliance on more costly computations" (p. 22).

- We thank the reviewer for prompting us to better explain how this task relates to the real world. We aim to manipulate and/or measure cues that we hypothesize are used in the real world to judge honesty. People hold beliefs about the statistical likelihood of events/stimuli. For example, an individual might have a sense of the probability of someone being a wealthy heiress or being as tall as 6'7". They can use these estimates to judge whether someone is truthful when they claim they are, for example, a wealthy heiress. The suggestion is not that statistical cues are provided in numerical form in real life, but rather that people have internal access to such rough estimates. Studies show these estimates (despite being noisy and sometimes biased) are on average not very far from the actual true statistics (for example, see Sharot et al., 2011). Similarly, for self-projection the notion is that people likely have a sense of their own likelihood to engage in a certain type of lie. For example, you may have a strong sense that the likelihood that you will present yourself as a wealthy heiress is close to zero, but the likelihood that you will say you liked a present which you did not is close to 90%. We now clarify this in the Discussion (page 24).

Rationality of the 'lying' behavior

Finally, I propose the authors point out that their experiments implemented an incentive structure in which participant can maximize income by always reporting the color 'blue'. This meant that in cases where the actual color was 'red', the income-maximizing response is to always say 'blue'.

- We added this in the Methods (page 26).

The question then is: do people infer that the experimenter has implemented an environment in which lying is the 'rational' thing to do? The experimenters also probed 'honesty' judgments. It thus seems that the experimenters' conveyed two conflicting messages as to what is the expected / desired behavior in the study. Can the authors say a bit more about what kind of assumptions / expectations individual respondents formed about the study, and how those may have informed their behaviors.

- We thank the reviewer for prompting us to say more about participants' expectations. First, the task creates a situation which mimics many real-life situations where lying is rational (i.e., the gain is high and the likelihood of punishment is low). Yet, people do not do so, potentially because many people

believe that lying is wrong (Abeler et al., 2019; Gerlach et al., 2019; Peterson, 1996).

- Second, at the end of the task, participants were asked questions that probed their assumptions and expectations about the task. We have now added a detailed analysis of participants' responses to the **Supplementary Materials** (pages 4-9). When probed about the purpose of the study, 57.1%-71.3% of participants across all experiments indicated they believed the study was about honesty, 19.4%-29.4% believed the study was unrelated to honesty and 7.4%-12.7% said they did not know (details in Supplementary Materials). We now show that whether they believed the study was about honesty, unrelated to honesty or they did not know, was unrelated to their mean suspicion (Experiment 1: $F(2) = 0.085$, $p = .919$; Experiment 2: $F(2) = 0.303$, $p = .739$; Experiment 3: $F(2) = 0.242$, $p = .786$) nor to their tendency to lie (Experiment 1: $F(2) = 0.078$, $p = .925$; Experiment 2: $F(2) = 0.432$, $p = .65$; Experiment 3: $F(2) = 0.727$, $p = .486$), nor to d' -scores (Experiment 1: $F(2) = 1.939$, $p = .149$; Experiment 2: $F(2) = 4.94$, $p = .009$; Experiment 3: $F(2) = 0.344$, $p = .709$).

Reviewer #3

The manuscript presents an investigation of the heuristics that people use to detect lies. In three experiments, the authors compared participants' heuristics to an optimal lie detector and found that participants assessed others' honesty based on their own honesty, whereas the optimal lie detector did not. In addition, they found that being more suspicious of others' lying leads to more lies of one's own, and people who lie more themselves are more suspicious of others' honesty.

The work presents an interesting and relevant research topic in the field and makes a valuable contribution to the literature.

- We thank the reviewer for this positive assessment.

However, I currently have some concerns regarding the task design and data analyses, which I elaborate on below.

1. The authors compared participants' heuristics to a statistical model trained using the actual feedback data (whether the player of interest actually lied or not). The statistical model relied on four features: the other player's behavior (lied or not), signed expectation violation, unsigned expectation violation, and who lost in the trial.

1a. The authors may have taken it too far by labeling this statistical model as an optimal lie detector, which suggests that no other model or strategy can surpass it in terms of lie detection. To establish its optimality, they need to demonstrate that this model outperforms all other reasonable candidate models, such as those incorporating additional features or employing non-linear combinations of the

features. Failing to provide such evidence may leave some readers skeptical about its optimality.

- Following the reviewer's comment, it became apparent that (i) we did not clearly explain the rationale behind this exercise and (ii) we should relabel the model. The point of the exercise was to test whether the cues human participants were using to assess honesty do in fact improve their judgements, or perhaps impair them. That is, are these cues associated with the ground truth, and would judgements be better if specific factors were not considered. As the reviewer notes, that exercise was not to identify other factors (beyond the ones tested on participants in this study) that may be optimal for lie detection. We now explain this in the Results and Methods (pages 8 and 29) and use the term "accurate lie detector model".

1b. The signed expectation violation is computed based on one's motivation to lie (positive if the motivation is to lie and negative otherwise), and the unsigned expectation violation. In fact, if I understand it correctly, it's lying motivation multiplied by the unsigned expectation violation, or in other words, the interaction between the two. I wonder if the authors have explored the possibility of including lying motivation as a dummy variable in their analyses and how that might affect the results. I am suggesting this because, typically, when a two-way interaction is included in a regression model, it is common to include both main effects as well.

To test if adding both main effects and the interaction in one model alters the significance of the beta coefficients for the interaction (signed expectation violation), we reran the Bayesian model averaging procedure with the suggested dummy variable. Both main effects and the interaction remained significant: (i) unsigned expectation violations (Experiment 1: weighted $\beta = 0.215$, 95%-CI = [0.189; 0.242]; Experiment 2: weighted $\beta = 0.212$, 95%-CI = [0.186; 0.238]; Experiment 3: weighted $\beta = 0.21$, 95%-CI = [0.184; 0.235]); (ii) The dummy variable (i.e., the other person's reported card colour; Experiment 1: weighted $\beta = 0.1$, 95%-CI = [0.088; 0.112]; Experiment 2: weighted $\beta = 0.114$, 95%-CI = [0.1; 0.128]; Experiment 3: weighted $\beta = 0.088$, 95%-CI = [0.077; 0.098]); (iii) signed expectation violation (Experiment 1: weighted $\beta = 0.254$, 95%-CI = [0.223; 0.286], Experiment 2: weighted $\beta = 0.226$, 95%-CI = [0.199; 0.254], Experiment 3: weighted $\beta = 0.226$, 95%-CI = [0.199; 0.254]). We now report this control analysis in the **Supplementary Materials** (page 15-16).

1c. How well did the optimal lie detector perform? I suggest that the authors report the accuracy and d'-scores of the detector and show us how much better the detector was compared to human judges. This will help readers evaluate the detector's performance.

- The accurate lie detector model's average accuracy was 0.806 (SD = 0.05) in Experiment 1, 0.777 (SD = 0.042) in Experiment 2 and 0.845 (SD = 0.036) in

Experiment 3. Its average d' -score was 1.44 (SD = 0.375) in Experiment 1, 1.56 (SD = 0.283) in Experiment 2 and 1.72 (SD = 0.312) in Experiment 3. Both these measures were larger than that of the human participant: accuracy in Experiment 1 (Wilcoxon signed rank test = 4596, $p < .001$), Experiment 2 (Wilcoxon signed rank test = 5535.5, $p < .001$) and Experiment 3 (Wilcoxon signed rank test = 4365.5, $p < .001$), and d' -scores in Experiment 1 (Wilcoxon signed rank test = 4922, $p < .001$), Experiment 2 (Wilcoxon signed rank test = 5595, $p < .001$) and Experiment 3 (Wilcoxon signed rank test = 4931, $p < .001$). We thank the reviewer for suggesting these analyses, which provide additional support for our conclusions (see pages 11, 16-17; 22-23 and Supplementary Materials page 12).

1d. One of the main conclusions is that the optimal lie detector does not take into account the opponent's behavior when assessing whether the player has lied. This conclusion appears evident and does not necessarily require statistical analyses. Given that the behaviors of both players were independent when conditioned on the shared presented card sets, it becomes apparent that, once controlling for the features of the card set that are related to lying probability (e.g., expectation violation and lying motivation), the opponent's behavior cannot be used to predict the other side's lying probability.

- The reason a statistical model may in principle use the behaviour of subject A to predict the behaviour of subject B (beyond expectation violation signed and unsigned) is because there may be a variable not captured by the other factors and not predicted by the experimenter that drive human lying and that factor is common to different participants. To give a nonsensical example, perhaps the order in which the cards are presented on screen is associated with lying. In such a case, one person's lying would be predictive of the other person's and this cue would be unassociated with the statistical likelihood already accounted for.

1e. Related to 1d, I suggest that the authors perform a more thorough comparison between the participants' heuristics and the optimal lie detector. For example, did the participants rely on the other features (e.g., signed/unsigned expectation violation) more or less than the optimal lie detector?

- Thank you for this suggestion. We compared the weight participants assigned to the cues to that of the accurate lie detector model. In all three experiments, humans relied on their own lying behaviour more than the accurate detector (non-overlapping CIs: Experiment 1 humans: mean weighted $\beta = 0.043$, 95%-CI = [0.033; 0.054] vs. accurate detector: mean weighted $\beta = 0$, 95%-CI = [0; 0]; Experiment 2 humans: mean weighted $\beta = 0.09$, 95%-CI = [0.068; 0.112] vs. accurate detector: mean weighted $\beta = 0$, 95%-CI = [0; 0]; Experiment 3 humans: mean weighted $\beta = 0.079$, 95%-CI = [0.06; 0.099] vs. accurate detector: mean weighted $\beta = 0$, 95%-CI = [0; 0]). They also put more weight on losing than the accurate lie detector (Experiment 1 humans: mean weighted $\beta = 0.053$, 95%-CI = [0.04; 0.066] vs. accurate detector: mean weighted $\beta = 0$, 95%-CI = [0; 0]; Experiment 2 humans: mean weighted $\beta = 0.05$, 95%-CI = [0.038; 0.062] vs.

accurate detector: mean weighted $\beta = 0$, 95%-CI = [0; 0]; Experiment 3 humans: mean weighted $\beta = 0.087$, 95%-CI = [0.065; 0.108] vs. accurate detector: mean weighted $\beta = 0.025$, 95%-CI = [0.019; 0.031]). In contrast, humans weighted unsigned expectation violations less than the accurate detector (Experiment 2 humans: mean weighted $\beta = 0.186$, 95%-CI = [0.141; 0.232] vs. accurate detector: mean weighted $\beta = 0.351$, 95%-CI = [0.267; 0.435]; Experiment 3 humans: mean weighted $\beta = 0.194$, 95%-CI = [0.146; 0.241] vs. accurate detector: mean weighted $\beta = 0.336$, 95%-CI = [0.282; 0.389]; trend in Experiment 1: humans: mean weighted $\beta = 0.197$, 95%-CI = [0.149; 0.245] vs. accurate detector: mean weighted $\beta = 0.313$, 95%-CI = [0.238; 0.387]). There was no difference between humans and the accurate lie detector regarding weights assigned to signed expectation violation (overlapping CIs: Experiment 1 humans: mean weighted $\beta = 0.354$, 95%-CI = [0.267; 0.44] vs. accurate detector: mean weighted $\beta = 0.295$, 95%-CI = [0.225; 0.365]; Experiment 2 humans: mean weighted $\beta = 0.345$, 95%-CI = [0.26; 0.429] vs. accurate detector: mean weighted $\beta = 0.329$, 95%-CI = [0.25; 0.408]; Experiment 3 humans: mean weighted $\beta = 0.31$, 95%-CI = [0.234; 0.385] vs. accurate detector: mean weighted $\beta = 0.305$, 95%-CI = [0.256; 0.353]). We now add these findings to the Results (pages 9, 14, 20).

1f. Although human judges relied on their own behaviors to assess whether the other side lied, the strength of this tendency appeared to be relatively weak, especially when compared to the influence of signed expectation violation and unsigned expectation violation (see Figures 2A, 5A, and 9A). Considering the weak relationship observed in the aggregate-level data, exploring the relationship at the individual level would be useful. Specifically, it would be useful to determine the proportion of participants who showed a 'significant' reliance on their own behaviors when judging others' honesty. If this proportion is small, the authors may need to modify their conclusions regarding the extent of humans' over-reliance on their own behaviors when judging others.

- Following the reviewer's comment, we now report the proportion of participants for whom each cue was significantly positive in the **Supplementary Materials** (pages 11 and 12) Indeed, all betas were significantly greater than zero for most participants in each experiment (lying oneself: 62.8%-71.6%; signed expectation violation: 89.2%-91.7%; unsigned expectation violation: 81%-86.1%; losing: 60.8%-67.8%).

Moreover, it remains unclear from the current discussion whether the aggregate-level results were primarily driven by individual differences (i.e., individuals who lied more also believed others lied more) or from trial-by-trial variations (i.e., people perceiving others as more dishonest if they themselves lied in a specific trial). Further clarification on this point would be useful.

- We find evidence on both aggregate and trial-by-trial levels across the three experiments that people who lie more are more suspicious of others (Experiment 1: standardised $\beta = 0.268$, $t(93) = 2.83$, $p = .006$; Experiment 2: standardised $\beta =$

0.381, $t(102) = 4.174$, $p < .001$, Experiment 3: standardised $\beta = 0.287$, $t(94) = 3.08$, $p = .003$) and that *when* they lie, they are more suspicious of others (Experiment 1: mean weighted $\beta = 0.043$, 95%-CI = [0.033; 0.054]; Experiment 2: mean weighted $\beta = 0.09$, 95%-CI = [0.068; 0.112]; Experiment 3: mean weighted $\beta = 0.079$, 95%-CI = [0.06; 0.099]). Because the relationship between lying oneself and suspicion appears in 62.8%-71.6% of the participants, it appears that the aggregate results are not primarily due to individual differences.

2. In experiment 1, in ~25% of the trials the participants observed the others lie, and in experiment 2, the rate was 50%. The authors found that participants lied more in experiment 2 than in experiment 1 and concluded that being more suspicious of others lying led to more lies of one's own. I'm a bit concerned by this conclusion because experiment 2 and experiment 1 were performed at different times, and thus several uncontrolled factors could have contributed to the observed difference in lying tendencies. To obtain more conclusive results, we need a single experiment, in which the percentage of trials where participants observe lying is manipulated within or between participants. I suggest the authors discuss this limitation, and state that their current conclusion is explorative.

- As suggested by the reviewer, we now note in the Discussion that Experiments 1 and 2 were not performed at the same time and thus conclusions that involve comparison between the two are tentative (page 24).

3. Minor comments

3a. On page 10, the authors wrote, “association between losing and suspicion was related to discernment (i.e., to d' -scores), due to the fact that when the other participant lies, they do so in their favor, which thus increases the likelihood that the participant will lose” to explain the positive association between betas for losing and lie detection d' -scores among participants. Could the authors elaborate on this point more? If this heuristic (i.e., losing or not) helps participants improve judgment accuracy, why does the optimal lie detector ignore this factor?

- The method used to detect the best fitting models (Bayesian averaging with BIC) uses ‘the parsimony principle’. That is, a simpler model with fewer parameters is favoured over more complex models with more parameters, provided the models fit the data similarly well. Thus, when ‘losing’ is not significant in the accurate detector model, it means that adding this parameter does not explain large enough additional variance on top of that already explained to justify its inclusion. In other words, while ‘losing’ can help explain accuracy, it does not necessarily do so beyond the information already considered by the accurate lie detector model. It is possible, however, that this cue is more salient for humans and thus explains more variance in their ratings.

3b. If possible, please include error bars for Figure 3a (and similar bar plots in the paper).

- We now include standard error bars for the regression coefficients in the updated Figures 3A, 5A and 8A.

3c. Have the authors tested how betas for different factors relate to prediction accuracy (rather than d'-scores)? Do all the conclusions stay the same?

- We now provide analyses on how betas for different cues relate to participants' detection accuracy (Supplementary Materials pages 12 and 13). All conclusions remain the same. Accuracy is not related to betas for participants' own lying behaviour (Experiment 1: standardized $\beta = 0.086$, $t(89) = 1$, $p = .319$; Experiment 2: standardized $\beta = 0.002$, $t(98) = 0.03$, $p = .973$; Experiment 3: standardized $\beta = -0.044$, $t(90) = -0.46$, $p = .649$), but is predicted by higher betas for statistical cues (i.e., signed expectation violation in Experiment 1: standardized $\beta = 0.303$, $t(89) = 2.92$, $p = .004$; Experiment 2 standardized $\beta = 0.715$, $t(98) = 8.64$, $p < .001$; Experiment 3: standardized $\beta = 0.212$, $t(90) = 2.07$, $p = .041$; and unsigned expectation violation in Experiment 1: standardized $\beta = 0.553$, $t(89) = 6.18$, $p < .001$; Experiment 2: standardized $\beta = 0.525$, $t(98) = 8.17$, $p < .001$; Experiment 3: standardized $\beta = 0.371$, $t(90) = 3.89$, $p < .001$). In other words, participants who rely more on statistical cues are better at discerning others' honesty.

3d. In experiment 2, how were the trials selected to fix "the proportion of trials with lies at 50%"?

- We randomly sampled 50% of the trials from the pool of past lies and 50% from the pool of past truths. We now clarify this in the Methods (page 27).

3e. On page 15 and page 19, the authors discussed the effect of gender, e.g., 'We also found an effect of gender on d'-scores (standardized $\beta = -0.147$, $t(98) = -2.325$, $p = .022$ '. I suggest the authors also comment on the direction of this result. Did male participants or female participants have higher d'-scores?

- Female participants had lower d'-scores in Experiments 2 and 3. We now note this in the Results (pages 15 and 20).

3f. In experiment 3, participants had different stakes in different blocks. Were they informed that the stakes for their opponents remained the same in different blocks? Otherwise, it seems sensible that some participants may assume that the other side would have the same stake as themselves and would base their honesty assessment on that.

We thank the reviewer for this comment. Participants were not informed about past participants' stakes, but may have assumed others had the same stakes as they did. To test if suspicion was driven by participants perceived stakes of others and/or by participants' own lying behaviour, we performed a mediation analysis. The mediation analysis examined if the relation between suspicion and block condition (high vs. low stakes) was mediated by whether participants lied. We found that

suspicion related to block condition (total effect = 0.026, $p < .001$, **Figure 6C**) and that this effect was partially mediated by when participants lied (indirect effect = 0.004, $p < .001$). Participants lied more in the high stakes block ($\beta = 0.044$, $p < .001$) and suspicions increased when participants lied ($\beta = 0.083$, $p < .001$). Thus, suspicion in Experiment 3 was indeed partially driven by when people lie. We add these findings to the Results (pages 17-18).

3g. On page 21, the authors wrote, “The key result is that greater reliance on one’s own behaviour (heuristic) led to more judgement errors, while reliance on statistical likelihoods and others’ motivation led to better discernment.” However, based on their analyses, there was no relationship between individual betas for lying oneself and lie detection d' -prime score. Therefore, reliance on one’s own behavior (heuristic) was not associated with (instead of positively correlated with) the number of judgment errors one made.

- Correct, this sentence was not accurately phrased in the past version of this manuscript. It should read “The key result is that reliance on self-projection of one’s own lying was not associated with discernment accuracy, while reliance on statistical likelihoods and inferred motivations led to better discernment”. We have now modified this in the Discussion (page 23). We thank the reviewer for pointing this out.

3h. What are the catch trials (page 23)? Please provide descriptions of how those trials were generated.

- We should have used the words “attention check trials” rather than “catch trials”. On attention check trials, participants were asked to select a specific option on the honesty rating scale (displayed horizontally) and submit the response (e.g., ‘Please select the third option from the left’). There was one every nine trials from trial 10 onward. This resulted in nine attention check trials in total per participant in Experiments 1 and 2, and six in total per participant in Experiment 3. We rectified this in the revised manuscript (see Methods, page 26).

31st Jan 24

Dear Professor Sharot,

Your manuscript titled "Why humans are bad at detecting lies" has now been seen by our reviewers, whose comments appear below. In light of their advice I am delighted to say that we are happy, in principle, to publish a suitably revised version in Communications Psychology under the open access CC BY license (Creative Commons Attribution v4.0 International License).

We therefore invite you to revise your paper one last time to address the remaining concerns of our reviewers and a list of editorial requests. At the same time we ask that you edit your manuscript to comply with our format requirements and to maximise the accessibility and therefore the impact of your work.

EDITORIAL REQUESTS:

SUBMISSION INFORMATION:

OPEN ACCESS:

Communications Psychology is a fully open access journal. Articles are made freely accessible on publication under a CC BY license (Creative Commons Attribution 4.0 International License). This license allows maximum dissemination and re-use of open access materials and is preferred by many research funding bodies.

For further information about article processing charges, open access funding, and advice and support from Nature Research, please visit <https://www.nature.com/commspsychol/article-processing-charges>

At acceptance, you will be provided with instructions for completing this CC BY license on behalf of all authors. This grants us the necessary permissions to publish your paper. Additionally, you will be asked to declare that all required third party permissions have been obtained, and to provide billing information in order to pay the article-processing charge (APC).

* TRANSPARENT PEER REVIEW: Communications Psychology uses a transparent peer review system. On author request, confidential information and data can be removed from the published reviewer

reports and rebuttal letters prior to publication. If you are concerned about the release of confidential data, please let us know specifically what information you would like to have removed. Please note that we cannot incorporate redactions for any other reasons.

* CODE AVAILABILITY: All Communications Psychology manuscripts must include a section titled "Code Availability" at the end of the methods section. We require that the custom analysis code supporting your conclusions is made available in a publicly accessible repository at this stage; please choose a repository that generates a digital object identifier (DOI) for the code; the link to the repository and the DOI must be included in the Code Availability statement. Publication as Supplementary Information will not suffice.

* DATA AVAILABILITY:

[link redacted]

Best regards,

Jennifer Bellingtier

Jennifer Bellingtier, PhD
Senior Editor
Communications Psychology

REVIEWERS' EXPERTISE:

Reviewer #1 Deception/(dis)honesty, decision making, computational modelling

Reviewer #2 Deception/(dis)honesty, decision making, computational modelling

Reviewer #3 Deception/(dis)honesty, decision making, computational modelling

REVIEWERS' COMMENTS:

Reviewer #1 (Remarks to the Author):

I thank the authors for their comprehensive revision and clear attempt to address the issues I raised in my first review. I think the current version is much improved. The overall tone of the conclusions is more balanced and aspects of the methods/results which were opaque in the first version are now

clearer. I appreciated the inclusion of the analysis related to the post-test questioning and the lens-model analysis in the supplementary materials.

There are still a few idiosyncrasies of interpretation, but I don't think these warrant further revision. These are aspects that future research can question/investigate.

I think this version can be published.

Reviewer #2 (Remarks to the Author):

I am ok with the way the authors addressed the issues I raised. I particularly appreciate the new analysis (SDT parameters and cross-validation) and the more coherent terminology throughout. I can recommend publication at this point.

Reviewer #3 (Remarks to the Author):

The authors have adequately addressed all of my feedback and made the necessary revisions to their paper. Congratulations on conducting a nice and informative study!